# $Chi^2$ weighted ensemble: A multi-layer ensemble approach for skin lesion classification using a novel framework - optimized RegNet synergy with Attention-Triplet

**Anwar Hossain Efat**[ID]*

Department of Computer Science and Engineering, IUBAT - International University of Business Agriculture and Technology, Dhaka, Bangladesh

* anwarhossainefat@gmail.com

**Data availability statement:** All data used in this study, including the augmented training data, are publicly accessible on the Kaggle repository: [HAM10000: Split and Augmented]

## Abstract

Skin lesions, including various abnormalities and potentially fatal skin cancers, require early detection for effective treatment. However, current methods often struggle to identify the precise areas responsible for these abnormalities after model dominance dispersion. To address this, we propose a novel Transfer Learning-based framework that integrates Optimized RegNet Synergy architectures and Attention-Triplet mechanisms—comprising channel attention, squeeze-excitation attention, and soft attention—combined with an advanced Ensemble Learning strategy. A significant gap in current research is the lack of techniques for optimal weight allocation in model predictions. Our study fills this gap by introducing the $Chi^2$ Weighted Ensemble (CWE) method, which is further enhanced into a Multi-Layer $Chi^2$ Weighted Ensemble (ML-CWE) to improve model aggregation across multiple layers. Evaluation on the HAM1000 dataset demonstrates that our ML-CWE approach achieves an impressive accuracy of 94.08%, outperforming existing state-of-the-art methods. To enhance model interpretability, we employ Gradient Class Activation Maps (Grad-CAM) to highlight critical regions of interest, improving both transparency and reliability. This work not only boosts accuracy but also facilitates early diagnosis, addressing challenges related to time, accessibility, and cost in skin lesion detection, and offering valuable insights for practical applications in dermatology.

## 1 Introduction

Skin lesions represent abnormal changes in the skin's appearance, while skin diseases encompass a broad range of conditions affecting the skin's health, structure, and function. These conditions range from common ailments like acne to more severe issues such as skin cancer. While skin diseases present a variety of symptoms, they are not solely defined by the presence of lesions. Lesions arise due to infections, inflammatory responses, allergic reactions, malignancies, insect bites, trauma, autoimmune disorders, genetic predispositions,

(https://www.kaggle.com/datasets/ahefatresearch/ham10000-split-and-augmented). Our use of the HAM10000 dataset complies with the Creative Commons Attribution-NonCommercial 4.0 International Public License because we have properly attributed the original dataset and cited the recommended paper by the authors. This fulfills the attribution requirement of the license. Additionally, our use of the dataset is for non-commercial purposes, aligning with the non-commercial clause of the license.

**Funding:** The author(s) received no specific funding for this work.

**Competing interests:** The authors have declared that no competing interests exist.

environmental factors, vascular anomalies, warts, and cysts, each with unique causes and characteristics [1]. Broadly, lesions are classified based on their potential harm. 'Benign skin lesions' are non-cancerous and generally pose no significant threat, including examples like moles, skin tags, warts, seborrheic keratoses, and hemangiomas. In contrast, 'malignant skin lesions' are cancerous, with the potential to spread to other parts of the body, with basal cell carcinoma, squamous cell carcinoma, and melanoma being the most common types [2].

Accurate diagnosis and effective treatment of skin conditions typically require a combination of clinical assessments and diagnostic tests. Neglecting symptoms leads to serious consequences, including the development of skin cancer, which is the most common form of cancer globally. Melanoma, although relatively rare, is the leading cause of skin cancer-related deaths [3]. According to recent estimates, around 2.2% of men and women are diagnosed with melanoma during their lifetime, with 97,610 new cases and 7,990 deaths reported in the United States in 2023. The impact of melanoma is significant, with over 1.4 million people currently living with the disease in the U.S [4].

Early detection of skin lesions is crucial for preventing the progression of serious conditions. However, many individuals are unaware of their skin abnormalities due to the extensive medical evaluations and associated costs. Dermatoscopy, or dermoscopy, is a non-invasive diagnostic technique that uses a magnifying device with lighting to examine skin lesions, aiding in the early detection of skin cancer and other skin conditions. While effective, the accuracy of dermatoscopy depends heavily on the expertise of the examiner, which introduces the potential for human error [5].

In contrast, AI-powered systems, particularly those leveraging Machine Learning (ML) and Deep Learning (DL) techniques, show great promise in skin lesion detection by enabling rapid image analysis, early diagnosis, and improved medical outcomes. Despite advancements by numerous researchers, challenges remain, including overreliance on data-rich classes and difficulties in capturing deep features in TL models without fine-tuning. Additionally, combining multiple models effectively is complex, and current models often lack interpretability while being prone to bias from using the same data for validation and testing. Transfer learning (TL) models like DenseNet and ResNet also face issues such as inefficient scaling, reliance on manual design choices, and rigidity in fixed scaling rules, which hinder adaptation to diverse environments and lead to suboptimal performance. High computational costs further limit their suitability for resource-constrained settings, highlighting the need for more flexible architectures and systematic exploration of design spaces to optimize performance across varied tasks [6].

In response to these challenges, researchers have explored the use of convolutional neural networks (CNNs) and ensemble learning techniques to overcome the limitations of individual models. However, the absence of an optimal weight selection process for each model in traditional ensemble techniques, such as majority voting, softmax averaging, and weighted averaging, limits the accuracy of results. These methods do not consider the varying importance of individual predictors, which leads to suboptimal outcomes. As a result, there is a clear need for more sophisticated approaches that effectively harness the strengths of different models while ensuring accurate and interpretable results.

Our methodology is strategically designed to directly confront these challenges, with the primary objective of addressing the following key research questions. These questions form the foundation for developing a robust architectural framework that provides well-informed responses.

**RQ1:** *What actions optimize the Transfer Learning model for specific tasks?*

- With a vast array of TL models available, selecting the appropriate one is challenging. Moreover, the fixed architecture of models trained on the ImageNet dataset may not be suitable for all tasks. Therefore, optimizing and tailoring a TL model to the specific task is crucial for achieving optimal performance.

**RQ2:** *What approach effectively identifies the most critical features, particularly significant areas or regions?*

- In classification tasks, not all regions of an image contribute equally to feature extraction; some introduce redundant or irrelevant information, negatively impacting results. Identifying and focusing on the most crucial regions is essential for improving classification accuracy.

**RQ3:** *Is a single algorithm sufficient, or is an Ensemble Learning (EL) technique necessary? If so, which should be employed?*

- No single algorithm consistently classifies all data accurately, making reliance on one method risky. Employing an EL technique mitigates this issue, offering a more reliable solution through a combination of algorithms.

**RQ4:** *What are the limitations of traditional EL methods that demand a new approach?*

- Traditional EL methods cannot always determine the optimal prediction ratio for each model in an ensemble. A novel approach that calculates and applies the optimal ratio for model predictions is necessary to enhance overall performance.

The above-mentioned inquiries are meticulously addressed, culminating in the study's significant contributions:

- The issue of class imbalance is effectively tackled through rigorous augmentation of the training dataset. This strategic augmentation ensures balanced class distribution, preventing model bias toward dominant classes. As a result, the architecture demonstrates reliability and impartiality in handling test and validation data.
- To optimize the TL model for specific tasks, we select the RegNet model due to its versatile design and advantages. We then optimize various versions of RegNet with customized layers and combine them into the "Optimized RegNet Synergy (ORNS)" network, capable of extracting both shallow and complex deep features.
- To focus on critical features, we ingeniously integrate Attention-Triplet (AT) mechanisms within customized architectures. This innovative approach ensures that models concentrate on the most essential aspects of the input data.
- We introduce a pioneering technique, the $Chi^2$ Weighted Ensemble (CWE), to enhance the model's robustness, accuracy, and generalization capabilities. This novel ensemble method operates across multiple layers, strategically boosting the architecture's performance.
- Grad-CAM visualization is incorporated to prioritize interpretability, allowing specific regions related to diagnosed skin conditions to be highlighted. This advanced visualization method enhances the transparency and insightfulness of the architecture.

The structure of the paper is meticulously organized to ensure clarity and coherence. It commences with an in-depth exploration of the existing literature in Sect 2, followed by a comprehensive presentation of the materials and methods in Sect 3. The subsequent Sect 4, provides a concise yet thorough analysis of the achieved performances. Building on these findings, Sect 5 engages in a comprehensive discourse, assessing the model's pragmatic implications along with minor improvement scopes, offering a holistic view. Ultimately, Sect 7 concludes the paper, encapsulating the essential takeaways and contributions of the study.

## 2 Literature review

The classification of skin lesions has been a well-explored area of research, with many studies contributing significantly to the understanding and advancement of this field. This section highlights the significant contributions of various researchers in this area. For instance, studies such as [7] through [11] have presented diverse approaches to classification and segmentation, each offering valuable insights for our current research. Additionally, a range of studies, from [12] to [17], have utilized Custom CNN architectures, while [16] and [17] have incorporated various transformation techniques. Innovative methods on skin lesion datasets have been explored in studies [18] and [19]. On the other hand, studies ranging from [20] to [27] have concentrated on feature extraction through TL, and studies [28] and [29] have combined soft attention mechanisms with TL.

Tajerian et al. [7] employed TL with EfficientNET-B1, achieving an 84.30% accuracy in identifying pigmented skin lesions. However, this approach struggled with highlighting specific features unique to skin datasets, affecting the diagnostic precision. In contrast, Hosny et al. [8] used AlexNet in a TL framework to classify skin lesions automatically, achieving high accuracy rates in diagnosing melanoma and nevus lesions. Dong et al. [9] introduced the TC-Net, a fusion network combining Transformer and CNN architectures, which significantly improved skin lesion segmentation by effectively integrating local and global features, outperforming other models like Swin UNet.

Khan et al. [10] developed the SkinViT architecture, incorporating an outlook attention mechanism, transformer blocks, and an MLP head block, which achieved up to 91.09% accuracy on different datasets, enhancing melanoma and nonmelanoma skin cancer classification. Singh et al. [11] proposed the SkiNet framework, which used Bayesian MultiResUNet for segmentation and DenseNet-169 for classification, achieving an accuracy of 86.67%, although this was considered suboptimal.

Saarela and Geogieva [12] proposed a Bayesian inference-based approach to improve model interpretability, achieving 80% accuracy, which was not particularly impressive. Sevli [13] created a CNN model for skin lesion classification, integrating it with a web application via a REST API and obtaining a 91.51% accuracy after evaluation by dermatologists. However, the custom CNN could not focus adequately on critical features. Shetty et al. [14] used a CNN to detect skin cancer, achieving a 94% accuracy, but their method was limited by using only a small subset of the dataset, raising concerns about generalizability.

Hoang et al. [15] used a lightweight neural network architecture, wide-ShuffleNet, for skin lesion classification, but it resulted in comparatively lower accuracy rates of 84.80% and 86.33% on different test datasets. Sun et al. [16] incorporated additional metadata and supplementary information during data augmentation, achieving an accuracy of 88.7% with a single model and 89.5% for the embedding solution, though the augmentation process lacked clarity. Nie et al. [17] presented a hybrid CNN-transformer model with focal loss, achieving an accuracy of 89.48%, but the approach had limitations in deep feature extraction.

Khan et al. [18] introduced a DL and Entropy-NDOELM-based architecture for multiclass skin lesion classification, fine-tuning EfficientNetB0 and DarkNet19 models, which achieved over 90% accuracy on all datasets. Ajmal et al. [19] developed a novel architecture combining DL models and a fuzzy entropy slime mould algorithm for feature optimization, achieving high accuracy on the HAM10000 and ISIC 2018 datasets with Grad-CAM for explainability.

Wang et al. [20] proposed a two-stream network combining DenseNet-121 and VGG-16, which extracted multiscale pathological information and achieved a 91.24% test accuracy, although the pre-trained model lacked fine-tuning. Mahbod et al. [21] explored the impact of image size on classification using TL, achieving a balanced multi-class accuracy of 86.2%,

although the model was heavy. Harangi et al. [22] used a TL-based CNN framework for multiclass classification with binary classification outcomes, achieving an average accuracy of 93.46%. However, the rationale for combining binary and multi-class classifications was not provided. Rahman et al. [23] created a weighted ensemble model using five deep neural networks via TL, enhancing the accuracy to 88%, but the model struggled with dataset specificity.

Popescu et al. [24] developed a skin lesion classification system using TL and collective intelligence, achieving an 86.71% validation accuracy through decision fusion but lacking independent test results. Nigar et al. [25] presented an explainable AI-based system using the LIME framework and ResNet-18, achieving 94.47% accuracy but relying on a small dataset and single pre-trained model. Gouda et al. [26] enhanced image quality using ESRGAN before applying a CNN, achieving 83.2% accuracy but not addressing data imbalance. Khan et al. [27] employed Resnet50 and a feature pyramid network for segmentation, followed by a 24-layer CNN for classification, achieving 86.5% accuracy but failing to utilize mask information during segmentation.

Datta et al. [28] explored the use of a soft-attention mechanism in skin cancer classification, achieving a 93.4% accuracy, though they struggled to find proper color channel weights. Nguyen et al. [29] combined DL with soft-attention, achieving accuracies of 90% and 86% with different models, but did not justify the choice of soft attention over other modules.

Building on the insights from these studies, our research addresses identified gaps by using the entire dataset and augmenting the training set to address data imbalance. This approach ensures the independence of the test set, providing a more accurate evaluation of the model on unseen data. We utilize the Attention Transfer (AT) method to identify crucial regions of interest and integrate them with TL models. Additionally, we fine-tune the TL models and ORNS architecture to reduce dependence on the ImageNet dataset, and our novel ensemble approach optimally weights predictions, overcoming previous limitations.

## 3 Materials and methods

### 3.1 Dataset description

Our study utilizes the publicly available Human Against Machine (HAM10000) dataset from the Harvard Dataverse repository, meticulously curated to encompass a diverse collection of skin lesion samples [30]. It includes 10,015 dermatoscopic images, all in jpg format, distributed into 7 classes: Melanoma (MEL), Nevus (NV), Vascular lesions (VASC), Actinic keratosis (AK), Basal Cell Carcinoma (BCC), Benign keratosis (BKL), and Dermatofibroma (DF), where MEL, AK, and BCC are types of cancer. NV, BKL, and DF are non-cancerous, whereas some types of VASC can be cancerous. The overview of the dataset is presented in Table 1.

In Fig 1, examples of images are displayed, with one sample provided per class in the dataset, while the high degree of class representation imbalance is corroborated by the class distribution depicted in Fig 2.

To align with the objectives of our study, the dataset is meticulously preprocessed. Details on the specific version used can be found in [31].

Table 1. Portrayal information of the dataset.

| No of images | Format | No of classes | Source |
|---|---|---|---|
| 10015 | JPG | 7 | Harvard Dataverse |

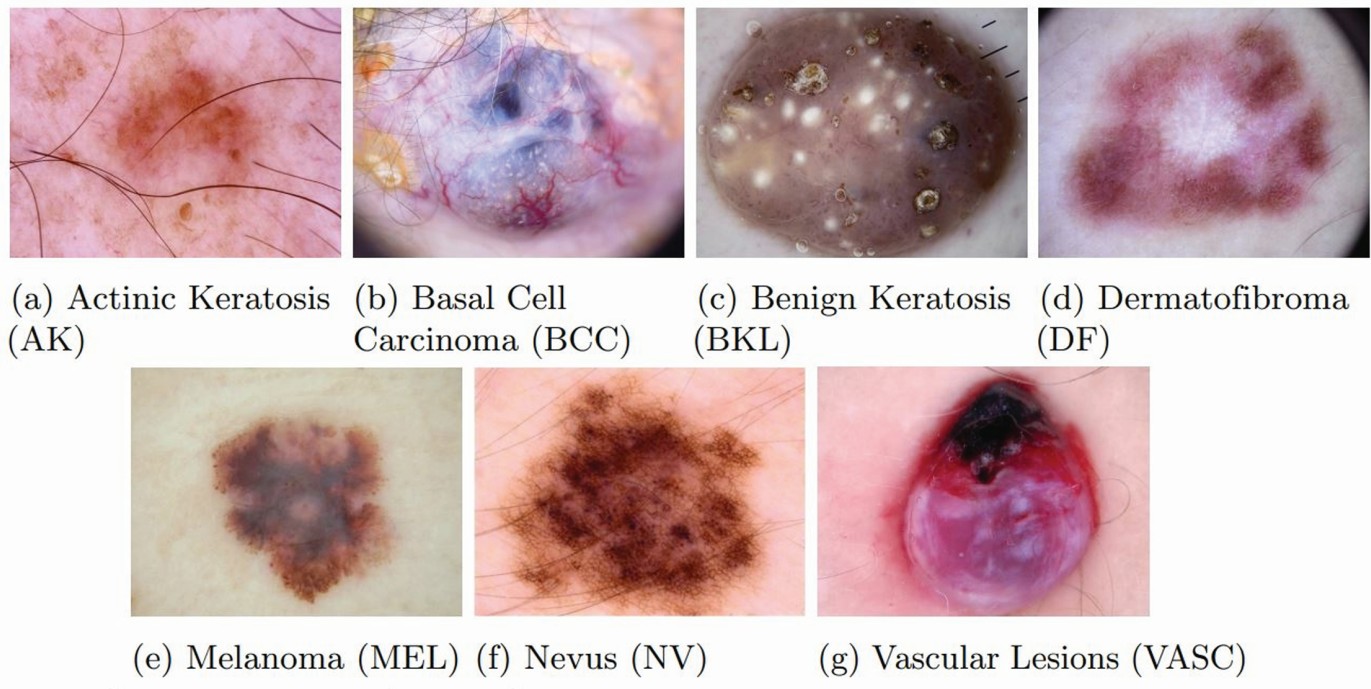

**Fig 1. Sample images of each class.**

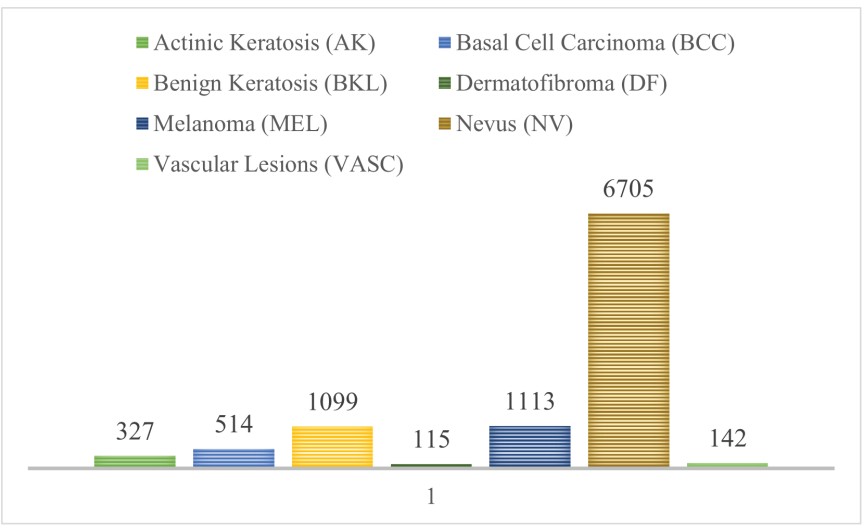

**Fig 2. Instance distribution for each class.**

## 3.2 Methodology

Our approach begins with dataset collection, followed by a vital phase of data preprocessing. The dataset is then partitioned into training, testing, and validation sets. To handle class imbalances, data augmentation is applied solely to the training set, ensuring that validation and testing remain unaffected by augmented data. We utilize Optimized RegNet Synergy

(ORNS) architectures, which are trained on the training data and validated on the validation data. These models are then evaluated using the testing set. Predictions from each architecture are integrated using the Multi-Layer $Chi^2$ Weighted Ensemble (ML-CWE) method to boost performance. The efficacy of CWE is assessed across several layers to support our conclusions. GradCAM visualization is ultimately employed to provide insight into the models' internal mechanisms. The sequential process of this investigation is illustrated in Fig 3.

### 3.3 Preprocessing and data augmentation

In this phase, we first categorize the images based on their lesion IDs and then carefully sample distinct images for the training, testing, and validation sets. We assign 15% of the images to both the testing and validation datasets, leaving 70% for the training dataset. To ensure that the testing set remains completely unseen during training, we introduce extra redundant images into the training set. This strategy enhances the robustness and reliability of our model by keeping the test data entirely separate from the training process. After this separation, we apply augmentation techniques exclusively to the training data, maintaining the independence of the test and validation sets. Through this approach, we generate around 8,000 images per class, addressing potential data imbalance issues effectively.

In our study, we implement a sophisticated image augmentation strategy using Tensor-Flow's 'ImageDataGenerator'. We begin by enhancing the contrast of the original images to improve their quality before augmentation. The augmentation process involves various transformations to significantly diversify the training data and strengthen the model's robustness. These transformations include random rotations up to 180 degrees, width and height shifts of 10%, and zoom variations within a 10% range. Additionally, horizontal and vertical flips are used to further increase variability. To handle any gaps introduced by these transformations, we use the nearest neighbor fill mode to ensure consistency in the augmented images. This comprehensive approach simulates a wide range of possible image variations, thereby enhancing the generalization ability of our DL model.

As shown in Fig 4, the original, contrast-enhanced, and augmented images are presented, using a sample from Vascular Lesions (VASC) along with its augmented versions. To address the issue of dataset imbalance, we aim to generate approximately 8,000 images in the training set for each class. Consequently, we achieve the following image distribution: AK (7,854), BCC (7,965), BKL (7,944), DF (7,377), MEL (7,932), NV (8,004), and VASC (7,706).

### 3.4 Creation of optimized RegNet synergy (ORNS) architectures

Our approach centers on the strategic utilization of Optimized RegNet models within the framework of Transfer Learning. Specifically, we focus on fine-tuning a diverse set of 24 pretrained models, including all 12 variants of both RegNetX (RNX) and RegNetY (RNY), which

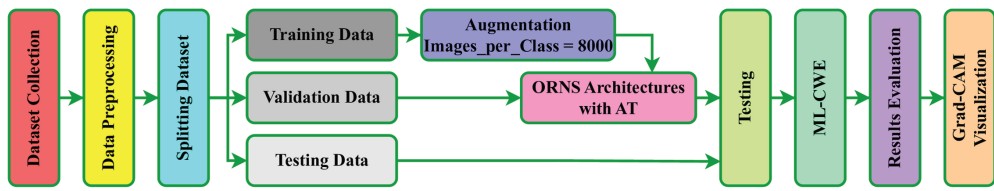

**Fig 3. Schematic representation of methodology.**

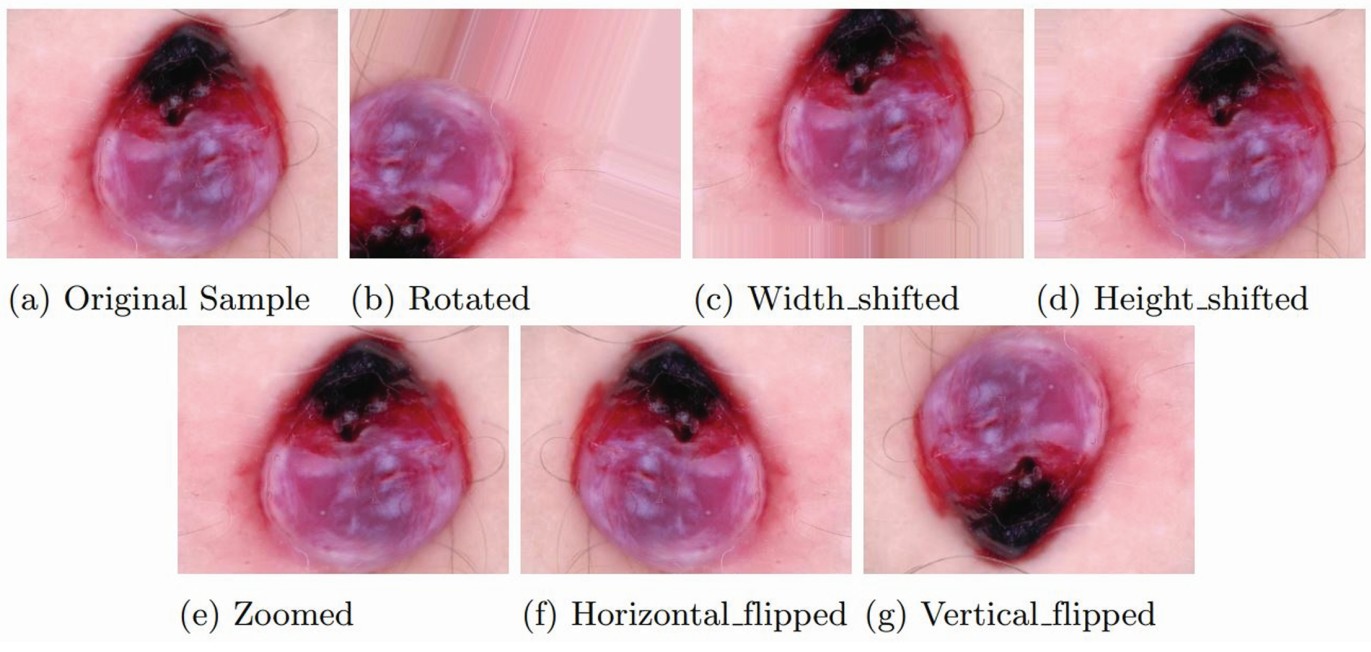

(a) Original Sample (b) Rotated (c) Width_shifted (d) Height_shifted

(e) Zoomed (f) Horizontal_flipped (g) Vertical_flipped

**Fig 4. Images of the augmented samples.**

accommodate input images of size 224x224x3. Recognizing that these models are not originally trained on our dataset, we meticulously fine-tune them to optimize the extraction of both shallow and deep features relevant to our data.

To achieve this, we introduce four customized CNN structures: Customized ORNS (C-ORNS), Channel Attention-based ORNS (CA-ORNS), Squeeze-Excitation Attention-based ORNS (SEA-ORNS), and Soft Attention-based ORNS (SA-ORNS). These architectures are specifically designed to leverage the power of AT, ensuring an enhanced focus on pertinent features during the learning process. The graphical depiction of the full architecture is illustrated in Fig 5.

The integration process begins by importing the pre-trained model from the 'keras' library, adapting it to our unique input shape, and transforming the output into a four-dimensional structure (None, height, width, channels). This adaptation aligns our model with the pre-trained one, allowing for seamless integration and effective feature extraction.

Fine-tuning is executed systematically, culminating in the generation of predictions from each individualized model for subsequent analysis.

The C-ORNS structure features two Convolution Blocks, each containing four 'Conv2D' layers with varying kernel sizes (7x7, 5x5, 3x3, 1x1), accompanied by 'BatchNormalization' layers. The first block employs 128 filters, while the second block uses 256 filters, with 'Max-Pooling2D' layers condensing the output. The ReLU activation function is consistently used across all convolutional layers, effectively mitigating vanishing gradient issues.

Again, we enhance the C-ORNS architecture by integrating a Channel Attention (CA) module within each convolution block. The CA layer is positioned after each 'Conv2D' layer and its corresponding 'BatchNormalization' layer, refining feature emphasis and selectively enhancing significant channel-wise information at intermediate stages of processing.

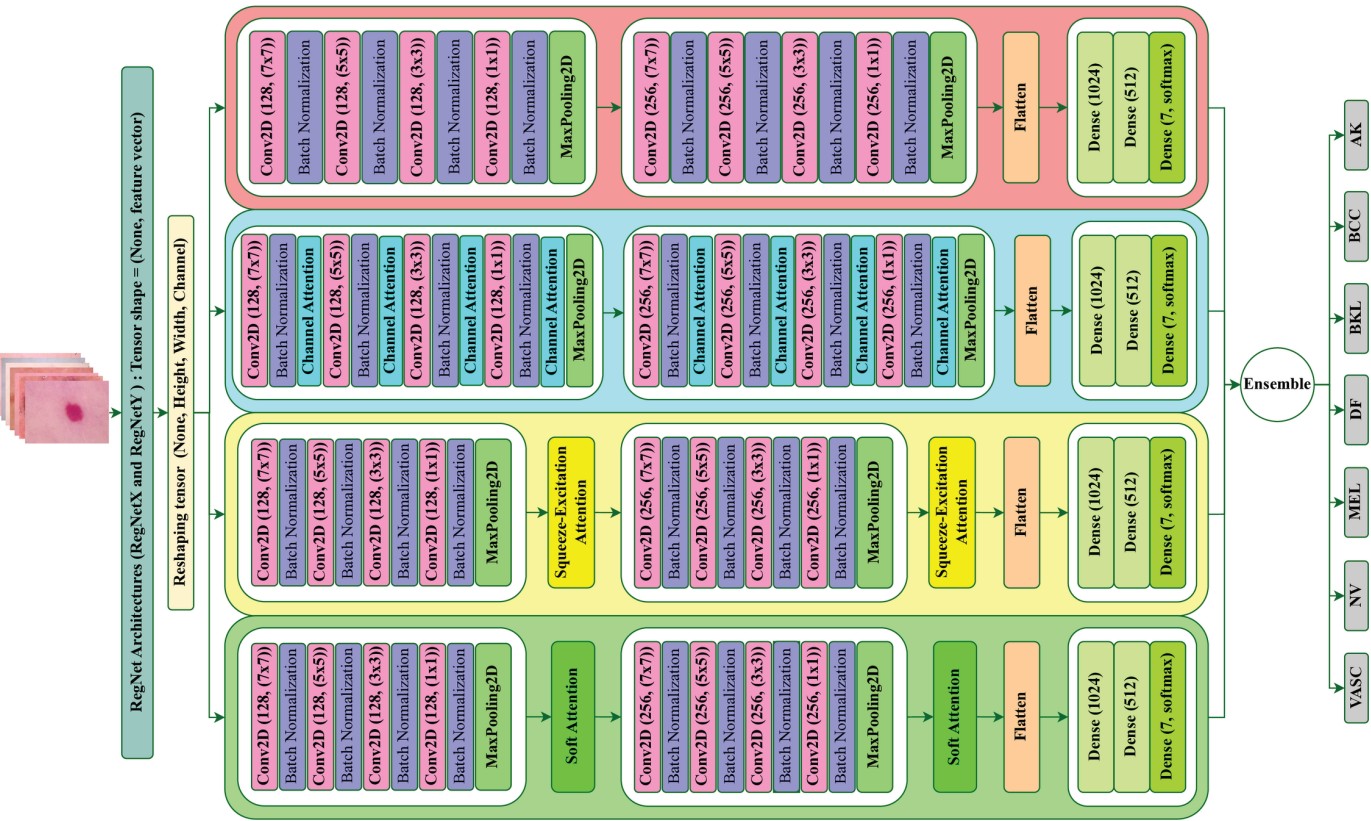

**Fig 5. ORNS architecture.**

In the SEA-ORNS variant, we embed the Squeeze-Excitation Attention (SEA) module after each convolution block, following the C-ORNS structure. The SEA layer recalibrates feature responses across channels, allowing high-level adjustment of channel importance and improving the model's ability to capture complex, hierarchical features.

For the Soft Attention (SA) module in SA-ORNS, we adopt a similar integration approach as in SEA-ORNS but position the SA layer after each convolution block rather than after every 'Conv2D' layer. This selective placement balances the increased parameter count while capturing fine-grained patterns within feature maps.

Finally, the output from the last max-pooling layer in each architecture is flattened and passed through three fully connected layers, configured with dimensions of 1024, 512, and 7 (corresponding to the number of classes). The first two layers utilize the ReLU activation function, while the final layer employs softmax activation to predict class probabilities.

This methodical architecture design ensures that our ORNS models are finely tuned and fully optimized for extracting the most relevant features from our dataset.

**3.4.1 Feature extraction process**   As previously mentioned, we employed ORNS models for effective feature extraction, excluding the top fully connected layers (using 'include_top=False') and applying global average pooling ('pooling='avg'). The resulting model output was reshaped to optimally supported dimensions, preparing it for further processing through multiple convolutional layers. These layers, equipped with filter sizes of 7x7, 5x5, 3x3, and 1x1, were systematically followed by ReLU activation and batch normalization to ensure stabilization and efficient learning. Max pooling layers were then utilized to reduce

spatial dimensions and sharpen feature focus. The extracted feature maps were eventually flattened and passed through a series of fully connected layers with ReLU activation, leading to a final dense layer with softmax activation for class probability prediction.

Fig 6 provides a visual representation of the feature map activations at various stages of the TL model, exemplified here by an Optimized RegNetX002 architecture. Each row corresponds to activations from different layers within the model, offering a step-by-step view of the feature extraction process:

- **Input Layer (input_1)**: Displays the preprocessed input image, showing the raw pixel data.
- **Zero Padding (zero_padding2d)**: Feature maps after zero padding, which prepares the input tensor for subsequent convolution operations.
- **Convolution (conv2d)**: Activation maps post convolution with 64 filters, revealing learned patterns and edges.
- **Batch Normalization (batch_normalization)**: Normalized feature maps following batch normalization, enhancing training stability and convergence.
- **ReLU Activation (activation)**: Output after the ReLU activation function, introducing the necessary non-linearity to the network.
- **Max Pooling (max_pooling2d)**: Downsampled feature maps post max pooling, reducing spatial dimensions while preserving critical features.
- **Concatenation (concatenate)**: Activation maps after concatenating feature maps from previous layers, integrating multi-path information.
- **Dense Layer (dense)**: Transformed feature maps into vector form before entering the fully connected dense layer.
- **Output Layer (dense_1)**: Final layer activations showing class probabilities through softmax activation.

Each subplot illustrates up to 5 filters per layer, using the 'viridis' colormap to ensure clarity and contrast. This figure offers valuable insights into how the model progressively processes and transforms input images, capturing hierarchical features crucial for accurate classification.

This approach, exemplified through a single sample and a subset of layers, underlines our strategy of extracting thousands of feature images. These images significantly contribute to enhancing the algorithm's overall performance by providing a detailed understanding of the feature extraction process across different layers.

**3.4.2 Attention-Triplet (AT)**   Our approach leverages a trio of attention modules, collectively referred to as the AT, to enhance model focus on critical input features while suppressing less relevant ones. This strategic incorporation of Channel Attention (CA), Squeeze-Excitation Attention (SEA), and Soft Attention (SA) enables our models to effectively capture and prioritize essential patterns within the data.

**3.4.3 Channel Attention (CA)**   The Channel Attention (CA) module refines feature maps by computing attention weights across channels. These weights, derived from the mean and standard deviation of the input feature maps, are applied to emphasize key features [32]. The process is mathematically represented as follows:

$$\mathbf{w_c} = \sigma\big(\mathbf{W_2}\delta\big(\mathbf{W_1}\mathbf{x}\big)\big), \tag{1}$$

$$\mathbf{y_c} = \mathbf{w_c} \odot \mathbf{x}, \tag{2}$$

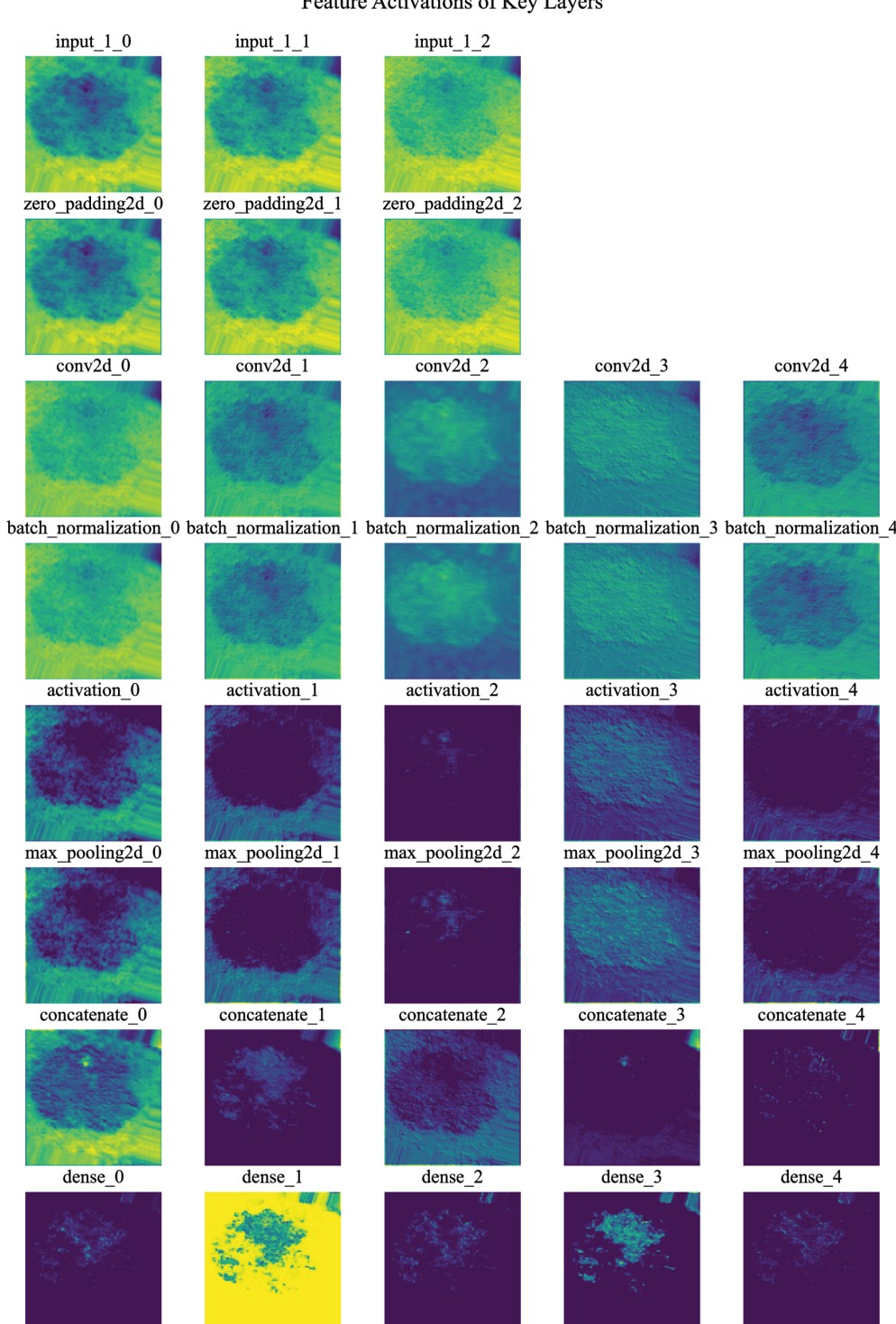

**Fig 6. Feature extraction after activation of each layer (one sample).**

where $\mathbf{x}$ denotes the input feature maps of size $C \times H \times W$, $\mathbf{W_1}$ and $\mathbf{W_2}$ are weight matrices, $\delta$ represents ReLU activation, $\sigma$ indicates sigmoid activation, $\mathbf{w_c}$ is the calculated channel attention weight, and $\odot$ denotes element-wise multiplication [32].

**3.4.4 Squeeze-Excitation Attention (SEA)** The Squeeze-Excitation Attention (SEA) module enhances the representational power of feature maps by combining spatial dimension reduction with channel-wise attention learning [33]. Given an input feature map $\mathbf{x}$ of size $C \times H \times W$, the SEA module operates as follows:

$$\mathbf{z} = \text{GlobalAvgPooling}(\mathbf{x}), \tag{3}$$

$$\mathbf{s} = \text{ReLU}(\mathbf{W}_2 \text{sigmoid}(\mathbf{W}_1 \mathbf{z})), \tag{4}$$

$$\mathbf{y} = \mathbf{s} \odot \mathbf{x}. \tag{5}$$

This approach ensures that the model dynamically recalibrates channel-wise feature responses, focusing on the most informative aspects of the input data.

**3.4.5 Soft Attention (SA)** The Soft Attention (SA) module assigns attention weights to individual input elements, prioritizing specific regions based on their importance [34]. The attention mechanism is expressed as:

$$a_i = \frac{\exp(e_i)}{\sum_{j=1}^{T} \exp(e_j)}, \tag{6}$$

where $a_i$ represents the attention weight for the $i$-th input element, $T$ is the input length, and $e_i$ is the scalar value associated with the $i$-th element [35].

This targeted emphasis enables the model to concentrate on the most relevant portions of the input, thereby improving overall performance.

## 3.5 $Chi^2$ weighted ensemble (CWE)

The novel approach of ensemble learning, CWE, is introduced by us. It calculates the most suitable weights for predictions from each classifier and then combines them through averaging, considering these weights. To achieve this, the concept of $Chi^2$ value is employed. The Chi-Square statistic plays a pivotal role in this approach by quantifying the accuracy of each classifier. The classifier with the highest $Chi^2$ value is considered the most accurate, as it exhibits the greatest alignment with the true label distribution. This method prioritizes classifiers that demonstrate a stronger relationship between predictions and actual outcomes, thereby enhancing the overall reliability of the ensemble. The sequential procedure for implementing CWE is outlined as follows.

### Step - 1

This method begins by assessing the performance of individual classifiers using the $Chi^2$ test. To facilitate this, correctly classified samples are labeled as class '1', while incorrectly classified ones are labeled as '0'. The $Chi^2$ test is a statistical measure used to evaluate the association between observed and expected frequencies in categorical data. It is calculated using the following formula:

$$\chi^2 = \sum \frac{(O_i - E_i)^2}{E_i}, \tag{7}$$

where $\chi^2$ represents the Chi-Square statistic; $O_i$ and $E_i$ denote the observed and expected frequencies, respectively. The $Chi^2$ value quantifies the discrepancy between observed and

expected frequencies, with a higher value indicating a stronger relationship between the classifier's predictions and the actual labels.

The observed frequencies are derived from the number of correctly and incorrectly classified samples for each classifier. Expected frequencies are calculated based on the overall distribution of the true labels. By applying the $Chi^2$ test, this method evaluates how well each classifier's predictions align with the expected distribution of labels. This serves as a performance indicator for each classifier.

### Step - 2

After obtaining the $Chi^2$ values for each classifier, the ensembling weights are computed. These weights are derived based on the ratio of each classifier's $Chi^2$ value to the total $Chi^2$ value across all classifiers. This ensures that classifiers with higher $Chi^2$ values contribute more prominently to the ensemble.

$$weight_i = \frac{\chi_i^2}{\sum_{i=1}^{n} \chi_i^2},\tag{8}$$

where $n$ is the number of classifiers.

### Step - 3

Finally, the predictions are averaged according to their respective weights, resulting in a blended output that leverages the strengths of each classifier within the ensemble. This can be achieved through the following process. Let $N$ be the number of individual classifiers in the ensemble. Each classifier $i$ produces predictions denoted as $P_i = [p_{i1}, p_{i2}, ..., p_{in}]$, where $n$ represents the number of instances in the dataset. The weights for each classifier, denoted as $w_i$, are determined based on the $Chi^2$ value. The ensemble prediction for each instance $j$ is calculated as:

$$E_j = \sum_{i=1}^{N} w_i \cdot p_{ij},\tag{9}$$

where $E_j$ is the final prediction for instance $j$, $w_i$ is the weight assigned to classifier $i$, and $p_{ij}$ is the prediction of classifier $i$ for instance $j$. The weights $w_i$ are determined based on the $Chi^2$ values, emphasizing the classifiers that exhibit better performance. Overall, the CWE technique combines individual classifiers' predictions using weighted averaging, where the weights are based on the Chi-Square statistic, enhancing the ensemble's accuracy and robustness. The overall approach of the CWE method is illustrated in Fig 7.

## 3.6 Multi-layer CWE

Our Multi-Layer CWE technique is applied across four distinct layers to effectively highlight and leverage the strengths of different models. This strategy is essential because, within

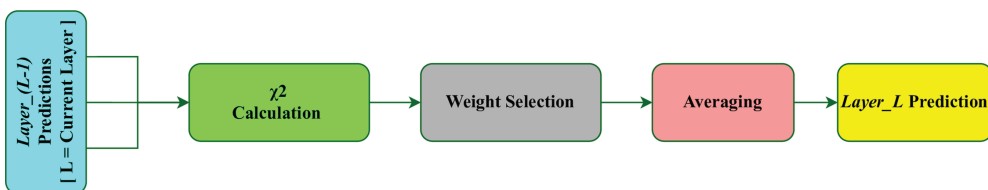

**Fig 7. $Chi^2$ weighted ensemble in layer - $L$.**

a single layer, assigning adequate importance to a superior model is challenging due to the relatively low individual classifier weights. To address this, we adopt a sequential "Layer by Layer" ensembling approach. This method allows us to progressively emphasize models that demonstrate superior performance at each layer, with their influence being further compounded as they are ensembled in subsequent layers. The "Layer by Layer" approach is detailed in the following sections, while a generic visual overview of the Multi-Layer CWE is provided in Fig 8.

**3.6.1 CWE in Layer 1**   In the first layer, we ensemble the predictions from the four foundational models—C-ORNS, CA-ORNS, SEA-ORNS, and SA-ORNS—for each classifier. This process results in a total of 24 predictions, derived from both RegNetX (RNX) and RegNetY (RNY), each containing 12 ORNS architectures.

**3.6.2 CWE in Layer 2**   The 24 predictions obtained from Layer 1 are then combined using two approaches. First, we separately ensemble all 12 architectures from both RNX and RNY, resulting in two distinct predictions. Second, we ensemble the common versions from both RNX and RNY to generate 12 additional predictions.

**3.6.3 CWE in Layer 3**   In Layer 3, we further ensemble the two predictions (RNX and RNY) obtained from Layer 2 to create a comprehensive third layer prediction (RNXY), which serves as the pre-final outcome. Additionally, the common versions' 12 ensembled predictions are combined to produce another key prediction, referred to as RN_XY.

**3.6.4 CWE in Layer 4**   Finally, the two predictions (RNXY and RN_XY) generated in Layer 3 are ensembled to form the ultimate prediction, denoted as RN. This final layer prediction represents the conclusive outcome of our study.

# 4 Experimental results and analysis

This section provides a comprehensive analysis, combining both theoretical insights and visual representations, to assess the classification performance. The primary objective of these results is to demonstrate the effectiveness of using CWE to improve the predictions of the ORNS architectures. By presenting experimental outcomes, which include a wide range of evaluation metrics, visual aids, and confusion matrices, we enable a detailed comparison of the different methods discussed earlier.

## 4.1 Performance evaluation metrics

To evaluate the performance and effectiveness of our models, we utilize several metrics such as accuracy, precision, recall (sensitivity), f1-score, specificity, and ROC-AUC (Receiver Operating Characteristic Area Under the Curve). These metrics provide valuable insights into the models' predictive capabilities. Each metric can be derived from the confusion matrix, which summarizes the model's predictions into true positives, false positives, true negatives, and false negatives. The mathematical expressions for these metrics are as follows:

$$Accuracy\ (Acc) = \frac{TP + TN}{TP + TN + FP + FN} \tag{10}$$

$$Precision\ (Pre) = \frac{TP}{TP + FP} \tag{11}$$

$$Recall\ (Re) = \frac{TP}{TP + FN} \tag{12}$$

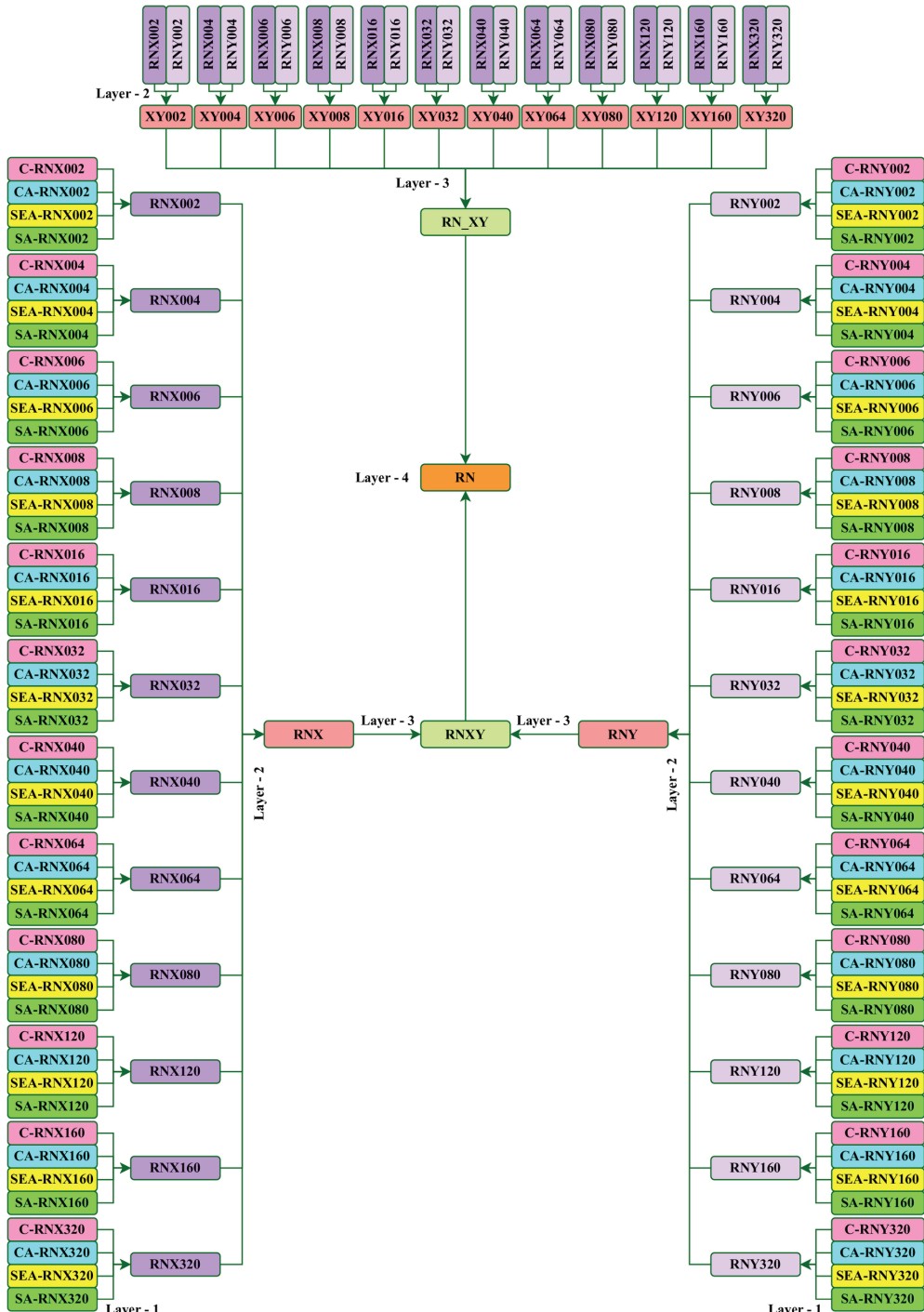

**Fig 8. Organization of multi-layer CWE.**

$$F1 - Score\ (F1) = 2 \times \frac{Precision \times Recall}{Precision + Recall} \tag{13}$$

$$Specificity\ (Spe) = \frac{TN}{TN + FP} \tag{14}$$

$$ROC\ (TPR) = \frac{TP}{TP + FN} \tag{15}$$

$$ROC\ (FPR) = \frac{FP}{FP + TN} \tag{16}$$

Where,

$TP$ : True Positives

$TN$ : True Negatives

$FP$ : False Positives

$FN$ : False Negatives

By thoroughly evaluating these metrics, we obtain a detailed understanding of our models' classification capabilities, allowing us to make informed decisions about their applicability in real-world scenarios.

## 4.2 Experimental setup

The entire architecture is implemented on a Kaggle notebook, utilizing a GPU P100 and a 2-core Intel Xeon CPU with a clock speed of 690 ms/step. The dataset, containing unique lesion images resized to (224, 224, 3), is split with 15% set aside for validation, another 15% for testing, and the remainder for training. The models are trained over 50 epochs with a batch size of 16. The Adam optimizer, initialized with a learning rate of 0.001, drives the optimization process. Categorical cross-entropy is used for loss calculation, with early stopping implemented through Reduce on Plateau with a patience of 25 epochs.

This section presents a thorough exploration of both theoretical concepts and graphical results to examine classification performance. The primary purpose of these findings is to validate CWE's effectiveness in enhancing model performance. By showcasing experimental results, including a broad array of evaluation metrics and graphical representations such as ROC-AUC curves and confusion matrices, we facilitate a robust comparison of the different approaches outlined in previous sections.

**4.2.1 Trainable parameters**   Since we ensemble the algorithms at prediction time, the number of trainable parameters remains unchanged post-ensemble. Table 2 summarizes the trainable parameters for each model.

It is evident that most of the RNY versions have a higher number of parameters compared to their RNX counterparts, with RNY320 and RNX320 having the highest parameter counts— SEA-ORNS trains approximately 154 million and 121 million parameters, respectively. The models ranked second have less than half of these parameters, and the remaining models have significantly fewer. However, since all algorithms operate independently and in parallel before being ensembled, the final prediction by CWE is achieved efficiently without introducing a significant time overhead.

**Table 2. Trainable parameters for each architecture.**

| Algorithms | Number of Trainable Parameters | | | |
|---|---|---|---|---|
| | C-ORNS | CA-ORNS | SEA-ORNS | SA-ORNS |
| RNX002 | 7,729,047 | 7,771,639 | 14,616,089 | 9,652,151 |
| RNX004 | 10,192,039 | 10,234,631 | 17,079,081 | 12,115,143 |
| RNX006 | 11,143,015 | 11,185,607 | 18,030,057 | 13,066,119 |
| RNX008 | 12,119,079 | 12,161,671 | 19,006,121 | 14,042,183 |
| RNX016 | 13,903,639 | 13,946,231 | 20,790,681 | 15,826,743 |
| RNX032 | 19,951,687 | 19,994,279 | 26,838,729 | 21,874,791 |
| RNX040 | 26,559,367 | 26,601,959 | 33,446,409 | 28,482,471 |
| RNX064 | 32,399,687 | 32,442,279 | 39,286,729 | 34,322,791 |
| RNX080 | 43,895,239 | 43,937,831 | 50,782,281 | 46,604,775 |
| RNX120 | 50,140,007 | 50,182,599 | 57,027,049 | 52,849,543 |
| RNX160 | 61,480,871 | 61,523,463 | 68,367,913 | 67,336,135 |
| RNX320 | 114,510,919 | 114,553,511 | 121,397,961 | 116,434,023 |
| RNY002 | 8,207,251 | 8,249,843 | 15,094,293 | 10,130,355 |
| RNY004 | 9,862,063 | 9,904,655 | 16,749,105 | 11,785,167 |
| RNY006 | 10,953,495 | 10,996,087 | 17,840,537 | 12,876,599 |
| RNY008 | 14,714,143 | 14,756,735 | 21,601,185 | 20,569,407 |
| RNY016 | 16,974,813 | 17,017,405 | 23,861,855 | 18,897,917 |
| RNY032 | 25,563,153 | 25,605,745 | 32,450,195 | 27,486,257 |
| RNY040 | 25,719,711 | 25,762,303 | 32,606,753 | 28,429,247 |
| RNY064 | 35,063,283 | 35,105,875 | 41,950,325 | 36,986,387 |
| RNY080 | 43,222,339 | 43,264,931 | 50,109,381 | 45,145,443 |
| RNY120 | 55,856,495 | 55,899,087 | 62,743,537 | 58,566,031 |
| RNY160 | 87,019,547 | 87,062,139 | 93,906,589 | 88,942,651 |
| RNY320 | 147,752,977 | 147,795,569 | 154,640,019 | 150,462,513 |

**4.2.2 Hyperparameters selection**  Hyperparameter tuning can significantly enhance performance, often in ways that exceed expectations [36]. The hyperparameters are carefully selected through a detailed process of manual tuning, guided by empirical observations and well-established practices in DL. Each aspect, from the learning rate and batch size to architectural decisions such as kernel sizes and activation functions, is meticulously assessed to enhance model performance and ensure resistance to overfitting. This approach is informed by extensive experimentation and a thorough understanding of the network's behavior, with the goal of balancing computational efficiency and achieving top-tier results for the task at hand.

We use a learning rate of 0.001 with the Adam optimizer to facilitate precise weight updates, which are essential for navigating the complex optimization landscape of our model. Batch normalization is incorporated to stabilize training by normalizing the inputs of each layer, thereby improving convergence speed and reducing the risk of overfitting. The 'he_normal' kernel initializer is employed to ensure effective weight initialization, aiding in the maintenance of gradient flow and enhancing the model's learning capacity. Additionally, the use of the 'ReLU' activation function allows our model to effectively capture complex patterns and relationships in the data, which is vital for achieving high accuracy in classification tasks.

## 4.3 Performance analysis of ORNS architectures in CWE

The CWE application with all classifiers at each layer, represented as $CWE_L$ (where $L$ stands for the layer), is employed to produce the $ML - CWE$.

**4.3.1 ORNS architectures in CWE-Layer 1** Twenty-four models, previously discussed, are utilized in combination with C-ORNS, CA-ORNS, SEA-ORNS, and SA-ORNS variants. The performance outcomes from these varied combinations, along with results from Layer-1 CWE, are summarized in Tables 3 through 14.

Table 3 presents the performance evaluation of the RNX002 and RNY002 models across different basic blocks, with CWE representing an ensemble of these blocks designed to harness their combined strengths. The CWE ensembles, RNX002 ($CWE_1$) and RNY002 ($CWE_1$), consistently outperform individual blocks.

For RNX002, the CWE ensemble achieves the highest accuracy at 91.43% and an F1-score of 90.85%, indicating that the ensemble approach yields a more robust and effective model compared to individual blocks.

In the same way, RNY002 ($CWE_1$) excels with an accuracy of 91.67% and an F1-score of 91.08%, outperforming all basic blocks. The ensemble's integration of attention mechanisms and customization leads to improved precision (91.13%) and recall (91.67%), underscoring its superior performance.

The CWE ensembles, by combining the strengths of the basic ORNS architectures, deliver superior performance across all key metrics, highlighting the value of this ensemble approach in optimizing model outcomes.

Table 4 depicts the performance evaluation of RNX004 and RNY004 models within CWE-Layer 1. The results highlight the effectiveness of the CWE ensemble approach compared to the individual basic blocks.

For RNX004, the CWE ensemble ($CWE_1$) achieves the highest performance across all key metrics, with an accuracy of 92.87% and an F1-score of 92.48%. These values surpass

**Table 3. Performance evaluation of RNX002 and RNY002 in CWE-Layer 1.**

| Algorithm | Accuracy | Precision | Recall | F1-score | Specificity |
|---|---|---|---|---|---|
| C-RNX002 | 89.49 | 88.78 | 89.49 | 89.01 | 82.18 |
| CA-RNX002 | 89.61 | 88.77 | 89.61 | 88.76 | 75.47 |
| SEA-RNX002 | 91.06 | 90.77 | 91.06 | 90.82 | 86.54 |
| SA-RNX002 | 89.86 | 88.96 | 89.86 | 89.13 | 81.73 |
| RNX002 ($CWE_1$) | 91.43 | 90.96 | 91.43 | 90.85 | 79.87 |
| C-RNY002 | 89.49 | 88.70 | 89.49 | 88.82 | 78.85 |
| CA-RNY002 | 89.37 | 88.71 | 89.37 | 88.76 | 78.79 |
| SEA-RNY002 | 90.22 | 89.35 | 90.22 | 89.49 | 77.42 |
| SA-RNY002 | 89.85 | 89.28 | 89.85 | 89.35 | 80.76 |
| RNY002 ($CWE_1$) | 91.67 | 91.13 | 91.67 | 91.08 | 79.40 |

**Table 4. Performance evaluation of RNX004 and RNY004 in CWE-Layer 1.**

| Algorithm | Accuracy | Precision | Recall | F1-score | Specificity |
|---|---|---|---|---|---|
| C-RNX004 | 91.55 | 91.29 | 91.55 | 91.21 | 85.64 |
| CA-RNX004 | 91.43 | 91.08 | 91.43 | 91.17 | 85.16 |
| SEA-RNX004 | 90.82 | 90.10 | 90.82 | 90.07 | 78.44 |
| SA-RNX004 | 90.46 | 90.83 | 90.46 | 90.51 | 87.02 |
| RNX004 ($CWE_1$) | 92.87 | 92.49 | 92.87 | 92.48 | 85.23 |
| C-RNY004 | 92.39 | 91.87 | 92.39 | 91.99 | 84.27 |
| CA-RNY004 | 91.43 | 91.08 | 91.43 | 91.19 | 86.14 |
| SEA-RNY004 | 90.58 | 89.68 | 90.58 | 89.86 | 78.91 |
| SA-RNY004 | 90.94 | 90.49 | 90.94 | 90.64 | 85.63 |
| RNY004 ($CWE_1$) | 92.63 | 92.08 | 92.63 | 92.20 | 84.28 |

those of the individual blocks, demonstrating the added value of the ensemble method. The CWE ensemble also shows a notable improvement in precision (92.49%) and recall (92.87%), indicating a more balanced model.

Likewise, the RNY004 model benefits significantly from the CWE ensemble, which reaches an accuracy of 92.63% and an F1-score of 92.20%. The precision (92.08%) and recall (92.63%) for the CWE variant also outperform the individual block variants, further confirming the advantages of combining these blocks.

Table 5 provides an assessment of the RNX006 and RNY006 models within CWE-Layer 1, highlighting the advantages of the CWE ensemble over individual basic blocks.

For RNX006, the CWE ensemble ($CWE_1$) surpasses individual blocks, achieving the highest accuracy at 92.39% and an F1-score of 91.87%. The ensemble also enhances precision (92.04%) and recall (92.39%), demonstrating the benefits of integrating the strengths of each block.

In a similar aspect, the RNY006 model benefits significantly from the CWE ensemble, achieving a peak accuracy of 92.63% and an F1-score of 92.23%, alongside high precision (92.28%) and recall (92.63%). These results underscore the ensemble's ability to deliver a more balanced and robust performance compared to the individual blocks.

Table 6 presents the performance evaluation of RNX008 and RNY008 models within CWE-Layer 1, highlighting the benefits of the CWE ensemble compared to individual basic blocks.

For RNX008, the CWE ensemble ($CWE_1$) achieves the highest performance across all metrics, with an accuracy of 93.60% and an F1-score of 93.30%. The ensemble also delivers strong

**Table 5. Performance evaluation of RNX006 and RNY006 in CWE-Layer 1.**

| Algorithm | Accuracy | Precision | Recall | F1-score | Specificity |
|---|---|---|---|---|---|
| C-RNX006 | 90.70 | 89.98 | 90.70 | 90.09 | 79.37 |
| CA-RNX006 | 91.91 | 91.70 | 91.91 | 91.49 | 85.17 |
| SEA-RNX006 | 90.94 | 90.35 | 90.94 | 90.35 | 80.82 |
| SA-RNX006 | 91.18 | 91.37 | 91.18 | 91.04 | 85.55 |
| RNX006 ($CWE_1$) | 92.39 | 92.04 | 92.39 | 91.87 | 83.74 |
| C-RNY006 | 90.94 | 90.16 | 90.94 | 90.27 | 80.34 |
| CA-RNY006 | 90.82 | 90.17 | 90.82 | 90.28 | 80.32 |
| SEA-RNY006 | 90.94 | 90.56 | 90.94 | 90.50 | 84.63 |
| SA-RNY006 | 92.03 | 91.84 | 92.03 | 91.82 | 87.12 |
| RNY006 ($CWE_1$) | 92.63 | 92.28 | 92.63 | 92.23 | 84.24 |

**Table 6. Performance evaluation of RNX008 and RNY008 in CWE-Layer 1.**

| Algorithm | Accuracy | Precision | Recall | F1-score | Specificity |
|---|---|---|---|---|---|
| C-RNX008 | 91.43 | 91.04 | 91.43 | 91.17 | 85.18 |
| CA-RNX008 | 91.91 | 91.67 | 91.91 | 91.63 | 85.69 |
| SEA-RNX008 | 92.39 | 91.87 | 92.39 | 91.93 | 82.80 |
| SA-RNX008 | 92.15 | 92.17 | 92.15 | 92.06 | 87.62 |
| RNX008 ($CWE_1$) | 93.60 | 93.29 | 93.60 | 93.30 | 86.24 |
| C-RNY008 | 90.10 | 89.23 | 90.10 | 89.31 | 76.50 |
| CA-RNY008 | 90.34 | 90.03 | 90.34 | 89.85 | 81.26 |
| SEA-RNY008 | 90.70 | 89.91 | 90.70 | 89.97 | 79.39 |
| SA-RNY008 | 92.03 | 91.45 | 92.03 | 91.51 | 80.89 |
| RNY008 ($CWE_1$) | 92.99 | 92.67 | 92.99 | 92.49 | 81.41 |

precision (93.29%) and recall (93.60%), outperforming the individual blocks and demonstrating the advantages of the ensemble approach.

Similarly, the RNY008 model shows significant performance improvements with the CWE ensemble. It achieves an accuracy of 92.99% and an F1-score of 92.49%, both of which are higher than those of the individual block variants. The CWE ensemble also exhibits enhanced precision (92.67%) and recall (92.99%).

Table 7 provides the performance evaluation of the RNX016 and RNY016 models within CWE-Layer 1, emphasizing the benefits of the CWE ensemble over individual basic blocks.

For RNX016, the CWE ensemble ($CWE_1$) stands out with the highest accuracy (93.36%) and F1-score (92.98%). These metrics surpass those of the individual blocks, illustrating the ensemble's superior ability to balance precision (92.97%) and recall (93.36%) effectively. This performance indicates that the CWE approach successfully combines the strengths of different blocks, leading to more robust results.

Similarly, RNY016 demonstrates notable performance improvements with the CWE ensemble. It achieves an accuracy of 93.24% and an F1-score of 92.80%, which are higher than the individual block variants. The ensemble's precision (92.83%) and recall (93.24%) further highlight its advantage in delivering balanced and enhanced model performance.

Table 8 highlights the comparative performance of RNX032 and RNY032 models in CWE-Layer 1, showcasing the benefits of the CWE ensemble over the individual basic blocks.

For RNX032, the CWE ensemble ($CWE_1$) stands out with an impressive accuracy of 92.87% and an F1-score of 92.39%, outperforming each individual block. The precision of 92.67% and recall of 92.87% underline the ensemble's capability to achieve a more comprehensive and balanced performance compared to the basic blocks.

**Table 7. Performance evaluation of RNX016 and RNY016 in CWE-Layer 1.**

| Algorithm | Accuracy | Precision | Recall | F1-score | Specificity |
|---|---|---|---|---|---|
| C-RNX016 | 92.51 | 91.85 | 92.51 | 91.96 | 84.75 |
| CA-RNX016 | 91.43 | 91.49 | 91.43 | 91.00 | 84.69 |
| SEA-RNX016 | 91.91 | 91.33 | 91.91 | 91.47 | 83.29 |
| SA-RNX016 | 92.15 | 92.22 | 92.15 | 91.99 | 89.02 |
| RNX016 ($CWE_1$) | 93.36 | 92.97 | 93.36 | 92.98 | 86.24 |
| C-RNY016 | 90.10 | 90.01 | 90.10 | 89.99 | 85.10 |
| CA-RNY016 | 91.30 | 91.12 | 91.30 | 90.84 | 84.64 |
| SEA-RNY016 | 92.75 | 93.04 | 92.75 | 92.78 | 91.42 |
| SA-RNY016 | 91.43 | 90.91 | 91.43 | 90.97 | 82.30 |
| RNY016 ($CWE_1$) | 93.24 | 92.83 | 93.24 | 92.80 | 85.26 |

**Table 8. Performance evaluation of RNX032 and RNY032 in CWE-Layer 1.**

| Algorithm | Accuracy | Precision | Recall | F1-score | Specificity |
|---|---|---|---|---|---|
| C-RNX032 | 91.18 | 90.36 | 91.18 | 90.49 | 81.34 |
| CA-RNX032 | 90.58 | 90.50 | 90.58 | 90.38 | 85.11 |
| SEA-RNX032 | 91.55 | 91.11 | 91.55 | 91.14 | 82.79 |
| SA-RNX032 | 90.34 | 90.51 | 90.34 | 90.32 | 86.48 |
| RNX032 ($CWE_1$) | 92.87 | 92.67 | 92.87 | 92.39 | 83.81 |
| C-RNY032 | 91.18 | 91.18 | 91.18 | 91.15 | 88.03 |
| CA-RNY032 | 92.03 | 92.07 | 92.03 | 91.78 | 85.61 |
| SEA-RNY032 | 90.46 | 91.12 | 90.46 | 90.18 | 86.95 |
| SA-RNY032 | 91.30 | 91.68 | 91.30 | 91.32 | 89.42 |
| RNY032 ($CWE_1$) | 93.48 | 93.47 | 93.48 | 93.22 | 87.16 |

In the case of RNY032, the CWE ensemble also excels, achieving the highest accuracy of 93.48% and an F1-score of 93.22%. The precision and recall rates of 93.47% and 93.48%, respectively, further demonstrate the ensemble's superior performance and effectiveness over the standalone models.

Table 9 presents the performance evaluation of RNX040 and RNY040 models within CWE-Layer 1, highlighting the advantages of the CWE ensemble over individual basic blocks.

For RNX040, the CWE ensemble ($CWE_1$) exhibits the highest performance with an accuracy of 93.12% and an F1-score of 92.59%. This outperforms the individual blocks, showcasing the ensemble's enhanced ability to deliver balanced and effective model results. The CWE ensemble also achieves superior precision (92.77%) and recall (93.12%), reinforcing its overall effectiveness.

In the case of RNY040, the CWE ensemble ($CWE_1$) shows improved results with an accuracy of 92.51% and an F1-score of 91.94%. Although it does not surpass all individual block metrics, it delivers strong performance in precision (92.16%) and recall (92.51%), demonstrating the ensemble's benefit in achieving high-quality results.

Table 10 presents the performance evaluation of the RNX064 and RNY064 models within CWE-Layer 1, emphasizing the CWE ensemble's effectiveness over individual basic blocks.

For RNX064, the CWE ensemble ($CWE_1$) stands out with the highest accuracy of 92.87% and an F1-score of 92.48%, surpassing the performance of individual blocks. This demonstrates the ensemble's superior ability to balance precision (92.45%) and recall (92.87%). Additionally, the CWE ensemble maintains a strong specificity of 83.33%, highlighting its overall effectiveness.

**Table 9. Performance evaluation of RNX040 and RNY040 in CWE-Layer 1.**

| Algorithm | Accuracy | Precision | Recall | F1-score | Specificity |
|---|---|---|---|---|---|
| C-RNX040 | 92.39 | 92.28 | 92.39 | 92.16 | 87.11 |
| CA-RNX040 | 92.15 | 91.54 | 92.15 | 91.69 | 84.75 |
| SEA-RNX040 | 91.30 | 90.85 | 91.30 | 90.53 | 83.67 |
| SA-RNX040 | 90.94 | 90.50 | 90.94 | 90.42 | 84.16 |
| RNX040 ($CWE_1$) | 93.12 | 92.77 | 93.12 | 92.59 | 83.80 |
| C-RNY040 | 92.27 | 92.06 | 92.27 | 91.84 | 83.30 |
| CA-RNY040 | 89.61 | 89.21 | 89.61 | 89.29 | 83.68 |
| SEA-RNY040 | 90.82 | 90.42 | 90.82 | 90.50 | 87.09 |
| SA-RNY040 | 91.18 | 90.64 | 91.18 | 90.77 | 83.75 |
| RNY040 ($CWE_1$) | 92.51 | 92.16 | 92.51 | 91.94 | 81.89 |

**Table 10. Performance evaluation of RNX064 and RNY064 in CWE-Layer 1.**

| Algorithm | Accuracy | Precision | Recall | F1-score | Specificity |
|---|---|---|---|---|---|
| C-RNX064 | 92.39 | 92.06 | 92.39 | 92.14 | 84.25 |
| CA-RNX064 | 91.67 | 91.26 | 91.67 | 91.23 | 81.87 |
| SEA-RNX064 | 91.43 | 90.85 | 91.43 | 91.01 | 83.26 |
| SA-RNX064 | 92.15 | 92.23 | 92.15 | 92.17 | 90.94 |
| RNX064 ($CWE_1$) | 92.87 | 92.45 | 92.87 | 92.48 | 83.33 |
| C-RNY064 | 92.15 | 91.84 | 92.15 | 91.95 | 88.09 |
| CA-RNY064 | 91.79 | 91.41 | 91.79 | 91.47 | 86.66 |
| SEA-RNY064 | 92.15 | 91.66 | 92.15 | 91.77 | 85.68 |
| SA-RNY064 | 93.12 | 92.72 | 93.12 | 92.65 | 87.19 |
| RNY064 ($CWE_1$) | 93.24 | 92.84 | 93.24 | 92.83 | 84.79 |

Similarly, the RNY064 model benefits significantly from the CWE ensemble, achieving the highest accuracy at 93.24% and an F1-score of 92.83%. The ensemble's precision (92.84%) and recall (93.24%) further emphasize its advantage in delivering a balanced and high-quality performance. The CWE ensemble also exhibits strong specificity at 84.79%, supporting its effectiveness in producing robust results.

Table 11 displays the performance evaluation of RNX080 and RNY080 models within CWE-Layer 1, demonstrating the clear advantages of the CWE ensemble over individual basic blocks.

For RNX080, the CWE ensemble ($CWE_1$) leads with the highest accuracy (93.72%) and F1-score (93.27%). It also achieves the best precision (93.40%) and recall (93.72%), surpassing all individual block variants. The specificity of 84.80% further emphasizes the CWE ensemble's robust performance across key metrics.

In the case of RNY080, the CWE ensemble also shows superior results, with an accuracy of 93.24% and an F1-score of 92.76%. The precision (92.78%) and recall (93.24%) are the highest among the tested variants, highlighting the ensemble's effectiveness in achieving balanced and high-quality performance. The specificity of 85.74% underscores its strong performance in distinguishing true negatives.

Table 12 presents the performance evaluation of RNX120 and RNY120 models within CWE-Layer 1, illustrating the advantages of using the CWE ensemble compared to individual basic blocks.

For RNX120, the CWE ensemble ($CWE_1$) achieves the highest accuracy (92.99%) and F1-score (92.43%). It also excels in precision (92.60%) and recall (92.99%), surpassing all individual basic blocks. The specificity of 80.47% indicates that the CWE ensemble maintains robust performance in distinguishing true negatives as well.

**Table 11. Performance evaluation of RNX080 and RNY080 in CWE-Layer 1.**

| Algorithm | Accuracy | Precision | Recall | F1-score | Specificity |
|---|---|---|---|---|---|
| C-RNX080 | 92.27 | 91.73 | 92.27 | 91.81 | 81.87 |
| CA-RNX080 | 91.79 | 91.25 | 91.79 | 91.35 | 84.22 |
| SEA-RNX080 | 91.91 | 91.29 | 91.91 | 91.38 | 81.85 |
| SA-RNX080 | 92.87 | 92.51 | 92.87 | 92.60 | 85.75 |
| RNX080 ($CWE_1$) | 93.72 | 93.40 | 93.72 | 93.27 | 84.80 |
| C-RNY080 | 92.75 | 92.27 | 92.75 | 92.22 | 85.24 |
| CA-RNY080 | 92.63 | 92.41 | 92.63 | 92.40 | 83.83 |
| SEA-RNY080 | 91.91 | 91.08 | 91.91 | 91.19 | 80.41 |
| SA-RNY080 | 92.51 | 92.16 | 92.51 | 92.21 | 89.52 |
| RNY080 ($CWE_1$) | 93.24 | 92.78 | 93.24 | 92.76 | 85.74 |

**Table 12. Performance evaluation of RNX120 and RNY120 in CWE-Layer 1.**

| Algorithm | Accuracy | Precision | Recall | F1-score | Specificity |
|---|---|---|---|---|---|
| C-RNX120 | 90.82 | 90.28 | 90.82 | 90.38 | 82.29 |
| CA-RNX120 | 91.30 | 91.01 | 91.30 | 91.04 | 84.69 |
| SEA-RNX120 | 92.03 | 91.43 | 92.03 | 91.23 | 77.54 |
| SA-RNX120 | 91.18 | 90.56 | 91.18 | 90.66 | 82.75 |
| RNX120 ($CWE_1$) | 92.99 | 92.60 | 92.99 | 92.43 | 80.47 |
| C-RNY120 | 91.55 | 91.02 | 91.55 | 91.03 | 85.16 |
| CA-RNY120 | 92.27 | 92.10 | 92.27 | 91.73 | 84.74 |
| SEA-RNY120 | 92.75 | 92.83 | 92.75 | 92.57 | 89.55 |
| SA-RNY120 | 92.51 | 92.11 | 92.51 | 92.18 | 85.68 |
| RNY120 ($CWE_1$) | 93.24 | 92.87 | 93.24 | 92.81 | 83.35 |

In a similar manner, for RNY120, the CWE ensemble ($CWE_1$) delivers top performance with an accuracy of 93.24% and an F1-score of 92.81%. It also shows the highest precision (92.87%) and recall (93.24%) among the variants, highlighting its effectiveness in achieving balanced and high-quality results. The specificity of 83.35% further supports the CWE ensemble's strong performance across key metrics.

Table 13 highlights the performance evaluation of the RNX160 and RNY160 models within CWE-Layer 1, underscoring the benefits of the CWE ensemble approach.

For RNX160, the CWE ensemble ($CWE_1$) surpasses the individual basic blocks, achieving the highest accuracy at 92.87% and an F1-score of 92.30%. It also excels in precision (92.40%) and recall (92.87%), demonstrating consistent and high-quality performance across these metrics. The specificity of 82.37% indicates its effectiveness in identifying true negatives.

Similarly, for RNY160, the CWE ensemble ($CWE_1$) achieves an accuracy of 92.63% and an F1-score of 92.15%. It shows competitive precision (92.02%) and recall (92.63%), with a specificity of 83.33%, further emphasizing the CWE ensemble's robust performance.

Table 14 presents the performance evaluation of RNX320 and RNY320 models in CWE-Layer 1, demonstrating the effectiveness of the CWE ensemble approach.

For RNX320, the CWE ensemble ($CWE_1$) achieves an accuracy of 93.36% and an F1-score of 93.00%. It excels in precision (93.08%) and recall (93.36%), indicating robust and consistent performance. The specificity is 87.16%, reflecting strong capability in identifying true negatives.

Likewise, for RNY320, the CWE ensemble ($CWE_1$) shows even higher performance with an accuracy of 93.60% and an F1-score of 93.20%. It also performs well in precision (93.18%) and recall (93.60%), with a specificity of 86.71%, underscoring its effective performance across all metrics.

**Table 13. Performance evaluation of RNX160 and RNY160 in CWE-Layer 1.**

| Algorithm | Accuracy | Precision | Recall | F1-score | Specificity |
|---|---|---|---|---|---|
| C-RNX160 | 91.06 | 90.30 | 91.06 | 90.41 | 80.83 |
| CA-RNX160 | 91.18 | 90.64 | 91.18 | 90.79 | 83.23 |
| SEA-RNX160 | 91.79 | 91.47 | 91.79 | 91.44 | 84.70 |
| SA-RNX160 | 91.79 | 91.29 | 91.79 | 91.26 | 84.23 |
| RNX160 ($CWE_1$) | 92.87 | 92.40 | 92.87 | 92.30 | 82.37 |
| C-RNY160 | 92.15 | 91.41 | 92.15 | 91.60 | 82.83 |
| CA-RNY160 | 92.87 | 92.45 | 92.87 | 92.55 | 88.61 |
| SEA-RNY160 | 91.79 | 91.14 | 91.79 | 91.23 | 79.92 |
| SA-RNY160 | 90.82 | 90.92 | 90.82 | 90.69 | 86.11 |
| RNY160 ($CWE_1$) | 92.63 | 92.02 | 92.63 | 92.15 | 83.33 |

**Table 14. Performance evaluation of RNX320 and RNY320 in CWE-Layer 1.**

| Algorithm | Accuracy | Precision | Recall | F1-score | Specificity |
|---|---|---|---|---|---|
| C-RNX320 | 90.34 | 89.83 | 90.34 | 89.94 | 83.19 |
| CA-RNX320 | 90.46 | 90.46 | 90.46 | 90.34 | 86.15 |
| SEA-RNX320 | 90.58 | 90.23 | 90.58 | 90.11 | 85.57 |
| SA-RNX320 | 91.43 | 91.63 | 91.43 | 91.42 | 90.38 |
| RNX320 ($CWE_1$) | 93.36 | 93.08 | 93.36 | 93.00 | 87.16 |
| C-RNY320 | 93.36 | 93.04 | 93.36 | 92.93 | 85.29 |
| CA-RNY320 | 92.75 | 92.30 | 92.75 | 92.32 | 85.25 |
| SEA-RNY320 | 91.30 | 90.50 | 91.30 | 90.72 | 82.30 |
| SA-RNY320 | 92.87 | 92.59 | 92.87 | 92.67 | 89.07 |
| RNY320 ($CWE_1$) | 93.60 | 93.18 | 93.60 | 93.20 | 86.71 |

Overall, in the initial layer of CWE, the ensembles consistently outperform individual basic blocks across various RegNet configurations, regardless of the model variant. This robust performance is observed across all key metrics, including accuracy, precision, recall, and F1-score, underscoring the effectiveness of the ensemble approach in optimizing model outcomes. By leveraging the strengths of both RNX and RNY architectures through ensembling, we achieve superior results, confirming the ensemble method's ability to enhance and maximize model performance across diverse scenarios.

**4.3.2 ORNS architectures in CWE-Layer 2** At Layer 2, our method utilizes two distinct approaches based on the predictions from Layer 1. Specifically, the first approach involves combining the twelve RNX models and twelve RNY models, resulting in two layer-2 predictions. The outcomes of this approach are detailed in Table 15. The second approach merges the common versions of both RegNet variants, producing twelve predictions. The results of this combined model are presented in Table 16.

Table 15 provides an insightful performance evaluation of the RNX and RNY models at Layer 2 within the CWE framework. The results clearly demonstrate the enhancement in model performance when utilizing the CWE ensemble approach compared to individual models.

For the RNX models, the CWE Layer 1 ($CWE_1$) already exhibits strong performance across the board, with accuracies ranging from 91.43% to 93.72%. Notably, the RNX080 and RNX320 models achieve the highest accuracies, highlighting the effectiveness of the CWE ensemble in aggregating predictions from multiple models. However, the Layer 2 ensemble ($CWE_2$) further refines these predictions, pushing the overall accuracy to an impressive 93.84%, along with superior precision (93.51%) and F1-score (93.41%).

**Table 15. Performance evaluation of RNX and RNY in CWE-Layer 2.**

| Algorithm | Accuracy | Precision | Recall | F1-score | Specificity |
|---|---|---|---|---|---|
| RNX002 ($CWE_1$) | 91.43 | 90.96 | 91.43 | 90.85 | 79.87 |
| RNX004 ($CWE_1$) | 92.87 | 92.49 | 92.87 | 92.48 | 85.23 |
| RNX006 ($CWE_1$) | 92.39 | 92.04 | 92.39 | 91.87 | 83.74 |
| RNX008 ($CWE_1$) | 93.60 | 93.29 | 93.60 | 93.30 | 86.24 |
| RNX016 ($CWE_1$) | 93.36 | 92.97 | 93.36 | 92.98 | 86.24 |
| RNX032 ($CWE_1$) | 92.87 | 92.67 | 92.87 | 92.39 | 83.81 |
| RNX040 ($CWE_1$) | 93.12 | 92.77 | 93.12 | 92.59 | 83.80 |
| RNX064 ($CWE_1$) | 92.87 | 92.45 | 92.87 | 92.48 | 83.33 |
| RNX080 ($CWE_1$) | 93.72 | 93.40 | 93.72 | 93.27 | 84.80 |
| RNX120 ($CWE_1$) | 92.99 | 92.60 | 92.99 | 92.43 | 80.47 |
| RNX160 ($CWE_1$) | 92.87 | 92.40 | 92.87 | 92.30 | 82.37 |
| RNX320 ($CWE_1$) | 93.36 | 93.08 | 93.36 | 93.00 | 87.16 |
| RNX ($CWE_2$) | 93.84 | 93.51 | 93.84 | 93.41 | 84.34 |
| RNY002 ($CWE_1$) | 91.67 | 91.13 | 91.67 | 91.08 | 79.40 |
| RNY004 ($CWE_1$) | 92.63 | 92.08 | 92.63 | 92.20 | 84.28 |
| RNY006 ($CWE_1$) | 92.63 | 92.28 | 92.63 | 92.23 | 84.24 |
| RNY008 ($CWE_1$) | 92.99 | 92.67 | 92.99 | 92.49 | 81.41 |
| RNY016 ($CWE_1$) | 93.24 | 92.83 | 93.24 | 92.80 | 85.26 |
| RNY032 ($CWE_1$) | 93.48 | 93.47 | 93.48 | 93.22 | 87.16 |
| RNY040 ($CWE_1$) | 92.51 | 92.16 | 92.51 | 91.94 | 81.89 |
| RNY064 ($CWE_1$) | 93.24 | 92.84 | 93.24 | 92.83 | 84.79 |
| RNY080 ($CWE_1$) | 93.24 | 92.78 | 93.24 | 92.76 | 85.74 |
| RNY120 ($CWE_1$) | 93.24 | 92.87 | 93.24 | 92.81 | 83.35 |
| RNY160 ($CWE_1$) | 92.63 | 92.02 | 92.63 | 92.15 | 83.33 |
| RNY320 ($CWE_1$) | 93.60 | 93.18 | 93.60 | 93.20 | 86.71 |
| RNY ($CWE_2$) | 93.72 | 93.40 | 93.72 | 93.27 | 84.80 |

Similarly, the RNY models under the CWE Layer 1 ($CWE_1$) demonstrate robust performance, with accuracies spanning from 91.67% to 93.60%. The RNY032 and RNY320 models stand out with the highest accuracy scores, underscoring the ensemble's ability to capture complex patterns. The Layer 2 ensemble ($CWE_2$) further consolidates these gains, achieving a near-perfect accuracy of 93.72%, accompanied by a precision of 93.40% and an F1-score of 93.27%.

Overall, the table illustrates the clear advantage of using a ML-CWE ensemble approach. The transition from Layer 1 to Layer 2 significantly enhances the performance metrics across all RNX and RNY models, making a compelling case for the effectiveness of this methodology in achieving superior predictive accuracy and reliability.

Table 16 presents the performance evaluation of the common variant ensemble of RNX and RNY architectures across twelve different model configurations at Layer 2. Each pair of RNX and RNY models is combined to create a unified prediction (denoted as 'XY'), showcasing the strength of the CWE ensemble at this layer.

For the smaller models, XY002 and XY004 deliver consistent performance, with accuracy values of 92.15% and 92.87%, respectively, reflecting the ensemble's reliability even in more compact configurations. Moving to the mid-range models, XY006, XY008, and XY016 also maintain strong results, with XY008 achieving the highest accuracy of 93.84% among these, demonstrating the ensemble's ability to leverage the strengths of both RNX and RNY.

The larger models, particularly XY032, XY064, and XY320, continue to exhibit robust performance, with XY320 reaching a peak accuracy of 93.84%, which matches the best results seen in the smaller models like XY008. This consistency across varying model sizes emphasizes the versatility and effectiveness of the CWE ensemble approach. Other models, such as XY040, XY080, XY120, and XY160, also perform well, with accuracy values ranging from 93.24% to 93.60%, further supporting the ensemble's overall effectiveness in combining the predictive strengths of both RNX and RNY variants.

**4.3.3 ORNS architectures in CWE-Layer 3** In this layer, two pre-final predictions are generated. The first prediction is derived from the ensemble of RNX and RNY models from the previous layer, referred to as RNXY in this layer. The second prediction comes from the ensemble of 12 common variants, labeled as RN_XY. The evaluations of these predictions are presented in Tables 17 and 18.

Table 17 captures the culminating evaluation of the RNXY architecture at CWE-Layer 3, showcasing the pre-final predictions derived from the ensemble of RNX and RNY models. More specifically, the RNXY ($CWE_3$) ensemble model demonstrates a slight but significant improvement across all metrics compared to the individual RNX and RNY ($CWE_2$) models. It achieves the highest accuracy at 93.96%, outstripping RNX ($CWE_2$) at 93.84% and RNY ($CWE_2$) at 93.72%. This model also leads in precision (93.64%), recall (93.96%), and F1-score (93.58%), confirming the effectiveness of this layered ensemble approach. The specificity of the RNXY ensemble is also enhanced to 85.30%, underscoring its robustness in distinguishing true negatives.

The incremental gains observed with the RNXY ($CWE_3$) model over its predecessors highlight the cumulative strength of the ensemble learning strategy employed throughout the layers. This strategic layering enhances predictive accuracy and consistency, culminating in a model that is not only superior in performance but also balanced across all evaluated metrics.

Table 18 provides a comprehensive evaluation of the ensemble method applied to twelve common variants of the RNX and RNY architectures, culminating in the RN_XY ensemble at CWE-Layer 3.

The individual XY models demonstrate strong performance metrics, with accuracy ranging from 92.15% for XY002 to 93.84% for models like XY008, XY080, and XY320. Notably, these

**Table 16. Performance evaluation of common variants ensemble of RNX and RNY in CWE-Layer 2.**

| Algorithm | Accuracy | Precision | Recall | F1-score | Specificity |
|---|---|---|---|---|---|
| RNX002 ($CWE_1$) | 91.43 | 90.96 | 91.43 | 90.85 | 79.87 |
| RNY002 ($CWE_1$) | 91.67 | 91.13 | 91.67 | 91.08 | 79.40 |
| XY002 ($CWE_2$) | 92.15 | 91.75 | 92.15 | 91.71 | 83.75 |
| RNX004 ($CWE_1$) | 92.87 | 92.49 | 92.87 | 92.48 | 85.23 |
| RNY004 ($CWE_1$) | 92.63 | 92.08 | 92.63 | 92.20 | 84.28 |
| XY004 ($CWE_2$) | 92.87 | 92.45 | 92.87 | 92.36 | 82.84 |
| RNX006 ($CWE_1$) | 92.39 | 92.04 | 92.39 | 91.87 | 83.74 |
| RNY006 ($CWE_1$) | 92.63 | 92.28 | 92.63 | 92.23 | 84.24 |
| XY006 ($CWE_2$) | 92.75 | 92.34 | 92.75 | 92.29 | 84.26 |
| RNX008 ($CWE_1$) | 93.60 | 93.29 | 93.60 | 93.30 | 86.24 |
| RNY008 ($CWE_1$) | 92.99 | 92.67 | 92.99 | 92.49 | 81.41 |
| XY008 ($CWE_2$) | 93.84 | 93.62 | 93.84 | 93.48 | 85.28 |
| RNX016 ($CWE_1$) | 93.36 | 92.97 | 93.36 | 92.98 | 86.24 |
| RNY016 ($CWE_1$) | 93.24 | 92.83 | 93.24 | 92.80 | 85.26 |
| XY016 ($CWE_2$) | 93.48 | 93.27 | 93.48 | 93.15 | 88.60 |
| RNX032 ($CWE_1$) | 92.87 | 92.67 | 92.87 | 92.39 | 83.81 |
| RNY032 ($CWE_1$) | 93.48 | 93.47 | 93.48 | 93.22 | 87.16 |
| XY032 ($CWE_2$) | 93.48 | 93.31 | 93.48 | 93.06 | 85.27 |
| RNX040 ($CWE_1$) | 93.12 | 92.77 | 93.12 | 92.59 | 83.80 |
| RNY040 ($CWE_1$) | 92.51 | 92.16 | 92.51 | 91.94 | 81.89 |
| XY040 ($CWE_2$) | 93.60 | 93.30 | 93.60 | 93.14 | 84.80 |
| RNX064 ($CWE_1$) | 92.87 | 92.45 | 92.87 | 92.48 | 83.33 |
| RNY064 ($CWE_1$) | 93.24 | 92.84 | 93.24 | 92.83 | 84.79 |
| XY064 ($CWE_2$) | 93.36 | 93.06 | 93.36 | 92.98 | 87.19 |
| RNX080 ($CWE_1$) | 93.72 | 93.40 | 93.72 | 93.27 | 84.80 |
| RNY080 ($CWE_1$) | 93.24 | 92.78 | 93.24 | 92.76 | 85.74 |
| XY080 ($CWE_2$) | 93.84 | 93.56 | 93.84 | 93.47 | 83.86 |
| RNX120 ($CWE_1$) | 92.99 | 92.60 | 92.99 | 92.43 | 80.47 |
| RNY120 ($CWE_1$) | 93.24 | 92.87 | 93.24 | 92.81 | 83.35 |
| XY120 ($CWE_2$) | 93.36 | 93.17 | 93.36 | 93.02 | 86.71 |
| RNX160 ($CWE_1$) | 92.87 | 92.40 | 92.87 | 92.30 | 82.37 |
| RNY160 ($CWE_1$) | 92.63 | 92.02 | 92.63 | 92.15 | 83.33 |
| XY160 ($CWE_2$) | 93.24 | 92.89 | 93.24 | 92.80 | 84.29 |
| RNX320 ($CWE_1$) | 93.36 | 93.08 | 93.36 | 93.00 | 87.16 |
| RNY320 ($CWE_1$) | 93.60 | 93.18 | 93.60 | 93.20 | 86.71 |
| XY320 ($CWE_2$) | 93.84 | 93.47 | 93.84 | 93.47 | 86.73 |

**Table 17. Performance evaluation of RNXY in CWE-Layer 3.**

| Algorithm | Accuracy | Precision | Recall | F1-score | Specificity |
|---|---|---|---|---|---|
| RNX ($CWE_2$) | 93.84 | 93.51 | 93.84 | 93.41 | 84.34 |
| RNY ($CWE_2$) | 93.72 | 93.40 | 93.72 | 93.27 | 84.80 |
| RNXY ($CWE_3$) | 93.96 | 93.64 | 93.96 | 93.58 | 85.30 |

models also exhibit consistent precision, recall, and F1-scores, underscoring their robustness across different configurations. For example, XY016 achieves a commendable F1-score of 93.15% and stands out with a specificity of 88.60%, the highest among the individual models, highlighting its balanced performance in both positive and negative classifications.

The RN_XY ensemble, however, elevates the predictive power to a new level, reaching an accuracy of 93.96%, with precision and recall closely aligned at 93.64% and 93.96%, respectively. The F1-score of 93.56% further reflects the ensemble's efficiency in synthesizing the strengths of the individual variants. Specificity remains competitive at 84.82%, demonstrating

**Table 18. Performance evaluation of RN_XY in CWE-Layer 3.**

| Algorithm | Accuracy | Precision | Recall | F1-score | Specificity |
|---|---|---|---|---|---|
| XY002 ($CWE_2$) | 92.15 | 91.75 | 92.15 | 91.71 | 83.75 |
| XY004 ($CWE_2$) | 92.87 | 92.45 | 92.87 | 92.36 | 82.84 |
| XY006 ($CWE_2$) | 92.75 | 92.34 | 92.75 | 92.29 | 84.26 |
| XY008 ($CWE_2$) | 93.84 | 93.62 | 93.84 | 93.48 | 85.28 |
| XY016 ($CWE_2$) | 93.48 | 93.27 | 93.48 | 93.15 | 88.60 |
| XY032 ($CWE_2$) | 93.48 | 93.31 | 93.48 | 93.06 | 85.27 |
| XY040 ($CWE_2$) | 93.60 | 93.30 | 93.60 | 93.14 | 84.80 |
| XY064 ($CWE_2$) | 93.36 | 93.06 | 93.36 | 92.98 | 87.19 |
| XY080 ($CWE_2$) | 93.84 | 93.56 | 93.84 | 93.47 | 83.86 |
| XY120 ($CWE_2$) | 93.36 | 93.17 | 93.36 | 93.02 | 86.71 |
| XY160 ($CWE_2$) | 93.24 | 92.89 | 93.24 | 92.80 | 84.29 |
| XY320 ($CWE_2$) | 93.84 | 93.47 | 93.84 | 93.47 | 86.73 |
| RN_XY ($CWE_3$) | 93.96 | 93.64 | 93.96 | 93.56 | 84.82 |

the ensemble's ability to maintain a high standard across various metrics. This layer represents the culmination of the model's iterative refinement, bringing together the best of each architecture into a final, high-performing ensemble.

**4.3.4 ORNS architectures in CWE-Layer 4** In the final evaluation at CWE-Layer 4, the performance metrics indicate a significant culmination of the ensemble strategies applied in the previous layers. Table 19 showcases the precision, accuracy, recall, F1-score, and specificity of the models, offering a comprehensive view of the refined prediction quality.

Both the RNXY and RN_XY models, derived from the earlier CWE-Layer 3 predictions, demonstrate identical accuracy (93.96%), precision (93.64%), and recall (93.96%). These models also maintain a high F1-score, with RNXY at 93.58% and RN_XY closely following at 93.56%. Specificity, a crucial metric for understanding the models' true negative rate, stands at 85.30% for RNXY and 84.82% for RN_XY, indicating robust performance in correctly identifying non-target classes.

The RN model, which represents the final amalgamation in CWE-Layer 4, slightly surpasses its predecessors with an accuracy of 94.08%, a precision of 93.77%, and a recall of 94.08%. The F1-score, crucial for balancing precision and recall, is also higher at 93.71%. Additionally, the RN model achieves a specificity of 85.78%, underscoring its enhanced ability to correctly exclude non-relevant instances.

This final result at CWE-Layer 4 marks the pinnacle of the ORNS architecture's performance, demonstrating the effectiveness of iterative refinement and the strategic combination of model predictions across layers.

## 4.4 Performance analysis by visualization

To simplify the analysis, we decide not to display confusion matrices for every classifier used, given the wide variety and differences among them. Instead, we focus on showing the

**Table 19. Performance evaluation of RN in CWE-Layer 4.**

| Algorithm | Accuracy | Precision | Recall | F1-score | Specificity |
|---|---|---|---|---|---|
| RNXY ($CWE_3$) | 93.96 | 93.64 | 93.96 | 93.58 | 85.30 |
| RN_XY ($CWE_3$) | 93.96 | 93.64 | 93.96 | 93.56 | 84.82 |
| RN ($CWE_4$) | 94.08 | 93.77 | 94.08 | 93.71 | 85.78 |

confusion matrices for the pre-final and final layers of the CWE model. These matrices, presented in Figs 9, 11, and 13, clearly illustrate the rates of correct and incorrect classifications for each category.

Similarly, we analyze the ROC-AUC curves, which are useful for visualizing a model's performance. We specifically focus on the ROC-AUC curves for the Multi-Layer CWE approach, similar to how we present the confusion matrices for only the last two layers. For consistency, these ROC-AUC curves are also shown in Figs 10, 12, and 14.

More specifically, the performance of our first pre-final architecture, RNXY ($CWE_3$), demonstrates its effectiveness across various classes. The VASC class achieves perfect accuracy, correctly classifying all 9 samples. In the DF class, 5 out of 6 samples are correctly identified, with only one misclassification. The NV class also performs impressively, correctly classifying 558 out of 663 samples, showing strong results for both minority and majority classes. For the AK class, 14 samples are correctly classified, with 8 errors, while the BCC class has 21 correct classifications and 6 mistakes. The BKL class performs well, with 52 out of 66 samples correctly classified. The MEL class, which is the most challenging, still manages 19 correct classifications out of 35 samples. Overall, the architecture shows high accuracy and robust performance across all classes.

The ROC curve AUC scores further highlight the strong performance of the RNXY ($CWE_3$) architecture. The MEL class, which has the lowest AUC score, still achieves an impressive 0.995, indicating excellent performance. The VASC class achieves a perfect AUC score of 1, while the other classes also perform well with AUC scores around 0.99. These minimal fluctuations in ROC curve demonstrate the architecture's precision and reliability.

Moving on to the performance of another pre-final architecture, RN_XY ($CWE_3$), we find that it also performs effectively across different classes, though slightly less so than RNXY. Starting with the most challenging class, MEL, we see that it successfully classifies 19 out of 35 samples. The BKL class performs well, with 53 out of 66 samples correctly classified. In the BCC class, 21 out of 27 samples are accurately classified, with 6 errors. The AK class identifies 13 samples correctly but has 9 misclassifications. The NV class showcases impressive performance, correctly classifying 558 out of 663 samples, indicating strong results in both minority

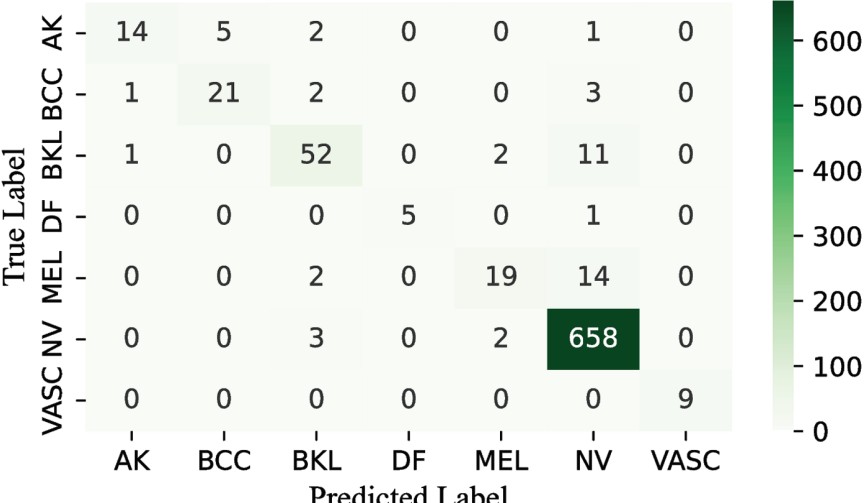

**Fig 9. Confusion matrix obtained by RNXY architecture in CWE-Layer 3.**

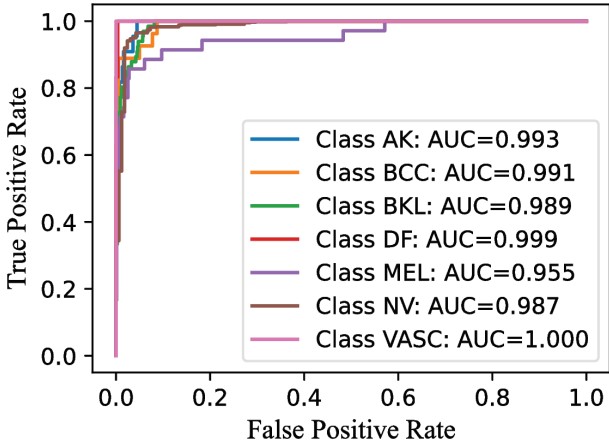

**Fig 10. ROC-AUC curve obtained by RNXY architecture in CWE-Layer 3.**

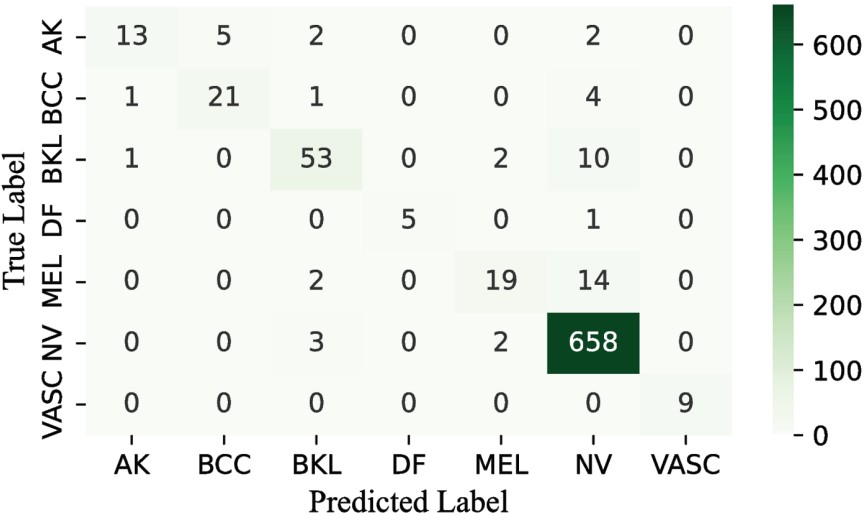

**Fig 11. Confusion matrix obtained by RN_XY architecture in CWE-Layer 3.**

and majority classes. For the DF class, 5 out of 6 samples are accurately classified, with only one misclassification. The VASC class achieves perfect accuracy, with all 9 samples correctly classified. Overall, while the RN_XY ($CWE_3$) architecture performs slightly less effectively than RNXY, it still demonstrates significant accuracy and robustness across all classes.

The ROC curve AUC scores for the RN_XY ($CWE_3$) architecture mirror those of RNXY, reinforcing its strong performance. The MEL class, despite having the lowest AUC score at 0.995, still shows excellent results. The AUC scores for the other classes also demonstrate very little fluctuation, underscoring the architecture's consistency and effectiveness.

Finally, the performance of our ultimate architecture, RN ($CWE_4$), stands out as the most effective across all classes, significantly outperforming previous layers. Starting with the VASC class, it achieves perfect accuracy with all 9 samples correctly classified, and an AUC score of 1, showcasing flawless performance. The DF class also performs exceptionally well, with

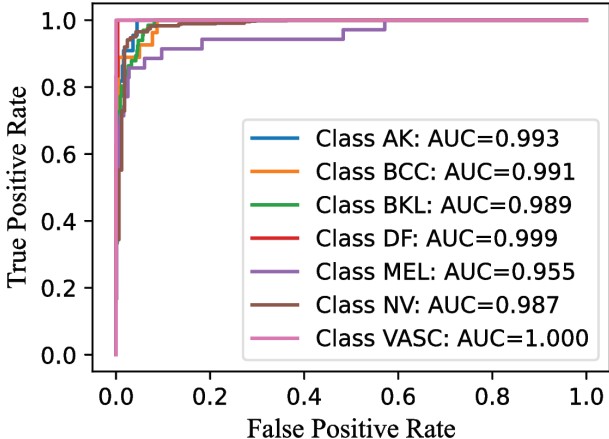

**Fig 12. ROC-AUC curve obtained by RN_XY architecture in CWE-Layer 3.**

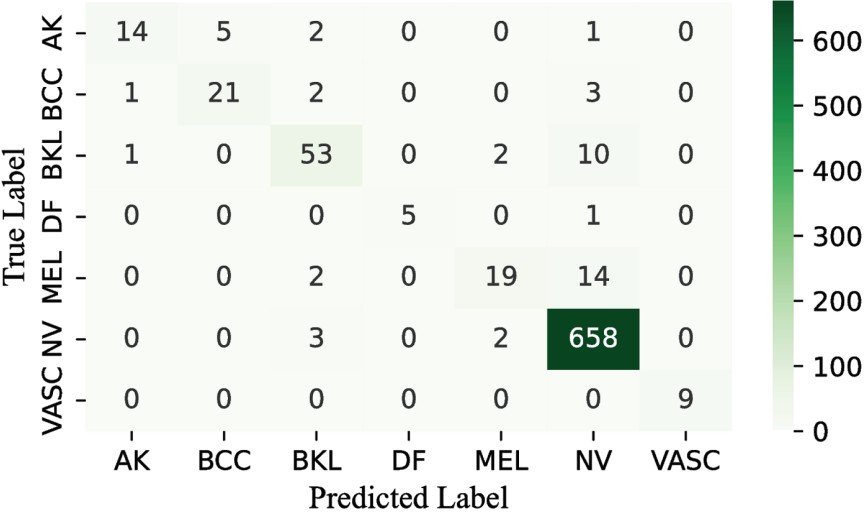

**Fig 13. Confusion matrix obtained by RN architecture in CWE-Layer 4.**

5 out of 6 samples accurately classified and only one misclassification, resulting in a near-perfect AUC score of 0.999. The AK class demonstrates strong results, correctly identifying 14 samples with 8 misclassifications, achieving an AUC score of 0.994, indicating good performance on the ROC curve. The NV class performs remarkably as well, correctly classifying 558 out of 663 samples and achieving an AUC score of 0.987, reflecting its capability in both minority and majority classes. Next, the BKL class manages to correctly classify 53 out of 66 samples, with an AUC score of 0.989, closely followed by the BCC class, which correctly classifies 21 out of 27 samples and attains an AUC score of 0.987. Lastly, the MEL class, which is the most challenging, still manages to classify 19 out of 35 samples correctly, achieving the lowest AUC score of 0.957. However, this score is still better than any obtained in the previous architectures. Overall, RN ($CWE_4$) showcases exceptional performance and robustness across all classes, with impressive accuracy and reliability.

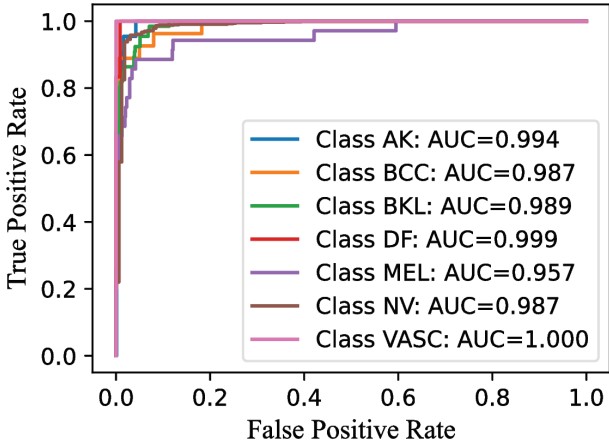

**Fig 14. ROC-AUC curve obtained by RN architecture in CWE-Layer 4.**

**4.4.1 Gradient Class Activation Map (GradCAM) for interpretability**   GradCAM (Gradient-weighted Class Activation Mapping) was employed as a visualization technique to improve the interpretability of the proposed model. It highlighted the regions in input images that contributed most significantly to the model's predictions. The final convolutional layer of the network was selected for generating activation maps, as this layer captures high-level spatial features crucial for classification.

The methodology for implementing GradCAM is depicted in Fig 15. The process began by constructing a gradient model capable of mapping input images to both the activations of the last convolutional layer and the model's final predictions. Using TensorFlow's GradientTape, the gradient of the class-specific output score with respect to the activations of the selected convolutional layer was computed.

These gradients were then spatially pooled by calculating the mean intensity for each feature map channel, which represented the importance of each channel for the target class. The pooled gradients were used to weight the activation maps of the final convolutional layer, and the weighted activations were aggregated to produce a class activation heatmap. This heatmap was normalized to a range between 0 and 1 for better visualization. Finally, the normalized heatmap was overlaid on the original input image using a colormap, highlighting the regions that influenced the model's decision.

This approach was extended to multiple classes by iterating over the test dataset and selecting representative images from each class. GradCAM outputs were generated for these images, revealing the regions of interest for each class. The resulting heatmaps demonstrated

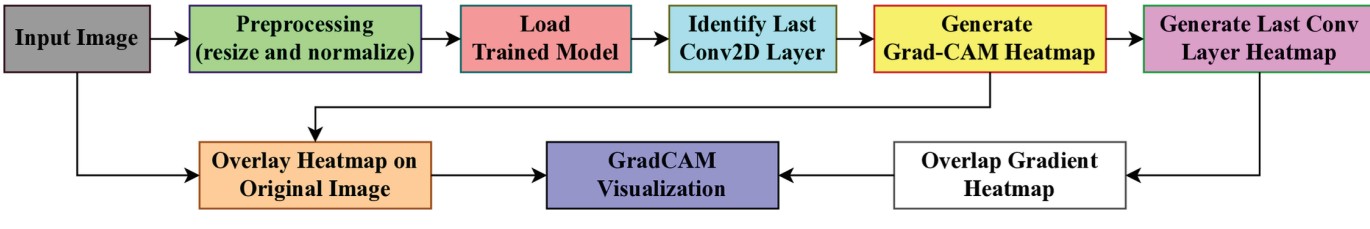

**Fig 15. Step by step implementation of gradient class activation map.**

the model's ability to focus on salient areas related to the target classes, such as lesions in medical images, thus confirming its capacity to identify key features.

Despite its strengths, GradCAM has certain limitations. Its reliance on model predictions means that any misclassification can result in inaccurate or misleading heatmaps. Additionally, for complex or subtle features, such as ambiguous skin lesions, GradCAM might sometimes highlight irrelevant regions, which can reduce its reliability in such cases.

This combination of interpretability and limitations emphasizes the importance of complementing GradCAM visualizations with robust model evaluation to ensure reliable insights.

In Fig 16, we present visualizations for each class, showcasing seven instances across seven different classes. The GradCAM views are seamlessly integrated with the original images, revealing how effectively our model focuses on the most critical regions rather than the entire image. This targeted approach substantially enhances the model's ability to classify images with higher accuracy, underscoring the strength of our contribution.

When a model successfully generates a precise heatmap that highlights the relevant region, it signals the model's ability to make accurate classifications. On the other hand, if the heatmap is misaligned, it often suggests that the model's classification may also be incorrect. To demonstrate this concept and evaluate the performance of AT, we provide an illustrative example in Fig 17 for clearer understanding.

In Fig 17, the original image serves as a representative example. Notably, by RNX002, both the CA and SA-based ORNS models successfully predict the correct class, while the SEA-based ORNS model does not. This pattern is observed across other models as well, reinforcing our argument. By integrating multiple models, our final prediction consistently proves to be accurate. This is clearly illustrated through the GradCAM visualizations of these models, which further strengthen our claim. The true innovation lies in our CWE ensemble approach, which enables precise class prediction by effectively compensating for the limitations of individual classifiers. This observation highlights the superiority of our advanced Multi-Layer CWE technique, demonstrating its ability to achieve accurate predictions where other methods may falter, thereby underscoring the strength of our contribution.

## 4.5 Ablation study

To highlight the superiority of our novel approach over state-of-the-art methods, we conduct an extensive ablation study centered on three key innovations: the significance of the AT, the superiority of the CWE, and the critical role of CWE across multiple layers. We assess the impact of these components by comparing the model's performance with and without each element, providing a clear analysis of their contributions.

**4.5.1 Utilization of CWE without AT** We apply CWE across various models, including all variants of RegNet, at different layers as previously mentioned. Each model is tested in four configurations: three with AT and one without AT. To highlight the efficacy of AT, we present the results in Table 20, showcasing the performance of CWE in Multi-Layer Ensemble (MLE) setup, excluding the AT-integrated models.

The table presents the performance metrics for the different configurations of the CWE model without AT, compared across various layers in the MLE setup. The analysis highlights that as the layers progress from $CWE_1$ to $CWE_4$, the performance generally improves, with the highest accuracy achieved by RN ($CWE_4$), which reports an accuracy of 93.84%. However, despite these improvements across layers, the absence of AT still limits the overall effectiveness of the models.

For instance, models like RNXY and RN_XY in $CWE_3$ achieve commendable results, with RNXY reaching an accuracy of 93.72%, and RN_XY achieving a similar accuracy of 93.72%.

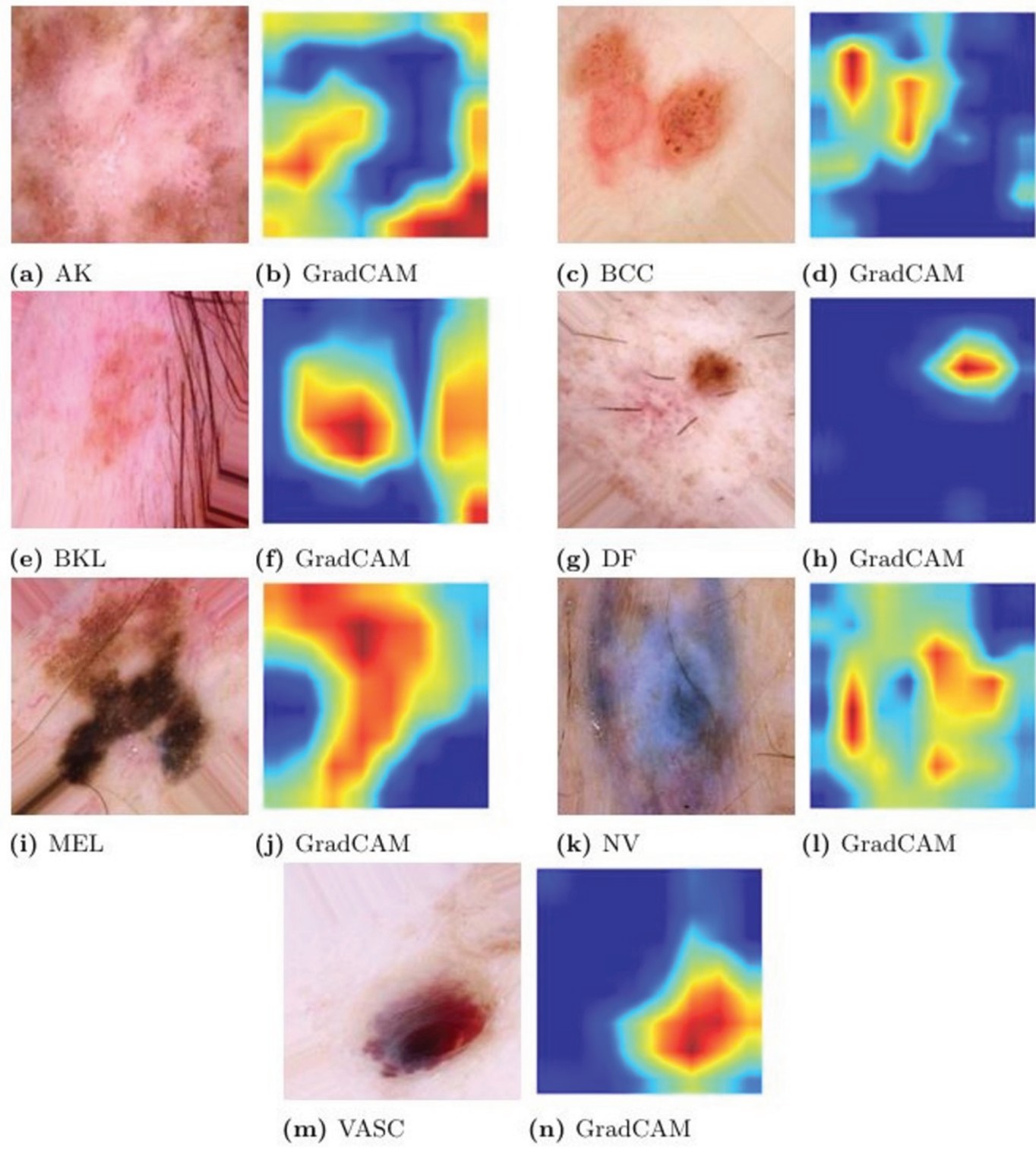

**Fig 16. GradCAM visualization for each class.**

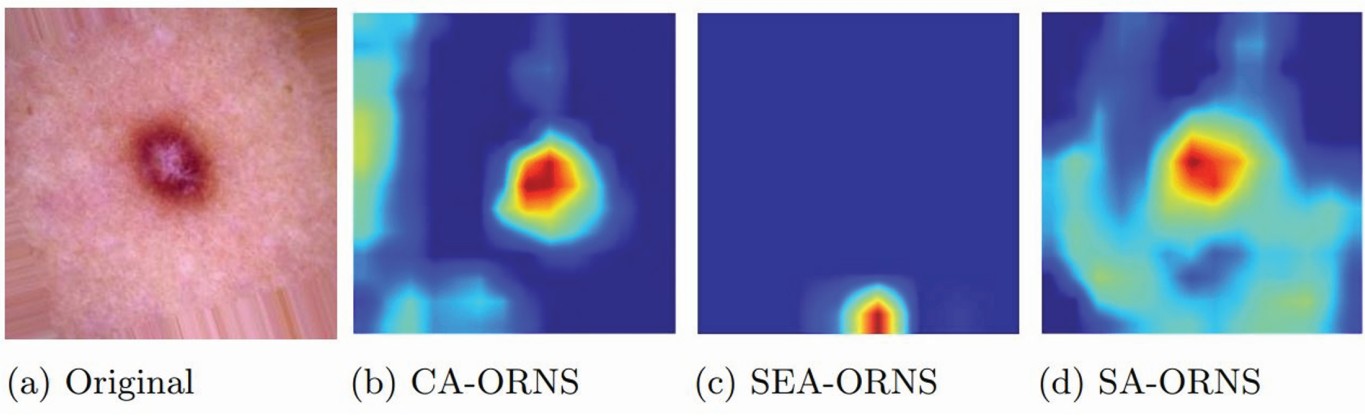

**Fig 17. GradCAM visualization for AT explainability (Example by RNX002).**

However, even the best-performing model without AT, RN ($CWE_4$), with its accuracy of 93.84%, precision of 93.54%, and F1-score of 93.42%, falls short compared to our proposed model.

Our proposed model, labeled as *Ours*, which utilizes AT, significantly surpasses all previous configurations. It achieves an accuracy of 94.08%, precision of 93.77%, and F1-score of 93.71%, clearly demonstrating the effectiveness of AT integration. This final model not only outperforms the configurations in $CWE_4$ but also proves the importance of AT in maximizing the model's potential.

**4.5.2 Utilization of conventional ensemble methods instead of CWE** As previously mentioned, we apply the CWE across multiple layers. Predictions from all architectures are combined by determining the optimal weights for each. To showcase the superiority of CWE, we compare its performance against traditional ensemble methods: Softmax Averaging (SA), Majority Voting (MV), and Weighted Averaging (WA) with random weights. In this comparison, each method applied at Layer $L$ is denoted as $Method_L$, such as $SA_L$ for Softmax Averaging. The results of these comparisons are presented in this section.

*4.5.2.1 Softmax Averaging (SA):*

As shown in the following tables, Softmax Averaging (SA) using the predictions at Layer $L$ is denoted as $SA_L$.

The results of SA across different layers are presented in Table 21, showcasing the performance metrics of accuracy, precision, recall, F1-score, and specificity. As expected, the performance of SA typically improves as we move up to higher layers. However, this trend is inconsistent, indicating a limitation in relying solely on SA for optimal ensemble outcomes.

For instance, while the $SA_1$ predictions for models such as RNX008 and RNX016 achieve accuracies of 93.60% and 93.36%, respectively, this improvement is not sustained across all layers. Specifically, at $SA_4$, the RN model achieves a relatively modest accuracy of 93.48%, falling short of expectations. Moreover, even the best performance metrics achieved by $SA_4$, precision of 93.16%, recall of 93.48%, F1-score of 93.02%, and specificity of 84.32%, do not surpass those of our proposed method, CWE, at Layer 4 ($CWE_4$). Our $CWE_4$ model outperforms SA, achieving a notable accuracy of 94.08%, precision of 93.77%, recall of 94.08%, F1-score of 93.71%, and specificity of 85.78%.

These results highlight the limitations of SA, particularly its inability to consistently improve performance across layers and its ultimate failure to outperform our CWE method. In contrast, CWE demonstrates superior performance, effectively leveraging the strengths of

**Table 20. Performance evaluation of CWE without AT.**

| Algorithm | Accuracy | Precision | Recall | F1-score | Specificity |
|---|---|---|---|---|---|
| C-RNX002 | 89.49 | 88.78 | 89.49 | 89.01 | 82.18 |
| C-RNX004 | 91.55 | 91.29 | 91.55 | 91.21 | 85.64 |
| C-RNX006 | 90.70 | 89.98 | 90.70 | 90.09 | 79.37 |
| C-RNX008 | 91.43 | 91.04 | 91.43 | 91.17 | 85.18 |
| C-RNX016 | 92.51 | 91.85 | 92.51 | 91.96 | 84.75 |
| C-RNX032 | 91.18 | 90.36 | 91.18 | 90.49 | 81.34 |
| C-RNX040 | 92.39 | 92.28 | 92.39 | 92.16 | 87.11 |
| C-RNX064 | 92.39 | 92.06 | 92.39 | 92.14 | 84.25 |
| C-RNX080 | 92.27 | 91.73 | 92.27 | 91.81 | 81.87 |
| C-RNX120 | 90.82 | 90.28 | 90.82 | 90.38 | 82.29 |
| C-RNX160 | 91.06 | 90.30 | 91.06 | 90.41 | 80.83 |
| C-RNX320 | 90.34 | 89.83 | 90.34 | 89.94 | 83.19 |
| RNX ($CWE_2$) | 93.48 | 93.11 | 93.48 | 93.05 | 83.84 |
| C-RNY002 | 89.49 | 88.70 | 89.49 | 88.82 | 78.85 |
| C-RNY004 | 92.39 | 91.87 | 92.39 | 91.99 | 84.27 |
| C-RNY006 | 90.94 | 90.16 | 90.94 | 90.27 | 80.34 |
| C-RNY008 | 90.10 | 89.23 | 90.10 | 89.31 | 76.50 |
| C-RNY016 | 90.10 | 90.01 | 90.10 | 89.99 | 85.10 |
| C-RNY032 | 91.18 | 91.18 | 91.18 | 91.15 | 88.03 |
| C-RNY040 | 92.27 | 92.06 | 92.27 | 91.84 | 83.30 |
| C-RNY064 | 92.15 | 91.84 | 92.15 | 91.95 | 88.09 |
| C-RNY080 | 92.75 | 92.27 | 92.75 | 92.22 | 85.24 |
| C-RNY120 | 91.55 | 91.02 | 91.55 | 91.03 | 85.16 |
| C-RNY160 | 92.15 | 91.41 | 92.15 | 91.60 | 82.83 |
| C-RNY320 | 93.36 | 93.04 | 93.36 | 92.93 | 85.29 |
| RNY ($CWE_2$) | 93.60 | 93.43 | 93.60 | 93.10 | 83.37 |
| RNXY ($CWE_3$) | 93.72 | 93.42 | 93.72 | 93.31 | 85.78 |
| XY002 ($CWE_2$) | 91.06 | 90.24 | 91.06 | 90.39 | 81.79 |
| XY004 ($CWE_2$) | 92.51 | 92.03 | 92.51 | 91.98 | 82.83 |
| XY006 ($CWE_2$) | 92.03 | 91.59 | 92.03 | 91.40 | 80.41 |
| XY008 ($CWE_2$) | 92.39 | 91.94 | 92.39 | 92.03 | 84.27 |
| XY016 ($CWE_2$) | 92.63 | 92.13 | 92.63 | 92.08 | 85.71 |
| XY032 ($CWE_2$) | 93.48 | 93.16 | 93.48 | 93.01 | 85.28 |
| XY040 ($CWE_2$) | 92.75 | 92.58 | 92.75 | 92.40 | 86.19 |
| XY064 ($CWE_2$) | 92.75 | 92.29 | 92.75 | 92.40 | 84.29 |
| XY080 ($CWE_2$) | 93.24 | 93.25 | 93.24 | 93.23 | 85.74 |
| XY120 ($CWE_2$) | 91.43 | 90.89 | 91.43 | 91.02 | 84.20 |
| XY160 ($CWE_2$) | 92.51 | 91.88 | 92.51 | 91.96 | 82.83 |
| XY320 ($CWE_2$) | 93.72 | 93.42 | 93.72 | 93.31 | 85.78 |
| RN_XY ($CWE_3$) | 93.72 | 93.39 | 93.72 | 93.26 | 83.85 |
| RN ($CWE_4$) | 93.84 | 93.54 | 93.84 | 93.42 | 83.86 |
| **Ours** | 94.08 | 93.77 | 94.08 | 93.71 | 85.78 |

multiple classifiers through optimal weighting, ultimately delivering the best results. Thus, it is evident that the accuracy obtained by $SA_4$ is 0.60% lower than that of our $CWE_4$.

*4.5.2.2 Majority Voting (MV):*

As shown in the upcoming table, Majority Voting (MV) using all predictions at Layer $L$ is denoted as $MV_L$.

The results for MV across different layers are presented in Table 22, showing how the ensemble method performs in terms of evaluation metrics as it progresses through the layers. As anticipated, the performance should ideally improve with higher layers, culminating in the final outcome at $MV_4$. However, the results reveal certain limitations of the MV approach.

**Table 21. Performance evaluation using SA instead of CWE.**

| Algorithm | Accuracy | Precision | Recall | F1-score | Specificity |
|---|---|---|---|---|---|
| RNX002 ($SA_1$) | 92.03 | 91.61 | 92.03 | 91.57 | 83.27 |
| RNX004 ($SA_1$) | 92.87 | 92.48 | 92.87 | 92.49 | 85.71 |
| RNX006 ($SA_1$) | 92.51 | 92.17 | 92.51 | 92.02 | 83.75 |
| RNX008 ($SA_1$) | 93.60 | 93.28 | 93.60 | 93.31 | 87.20 |
| RNX016 ($SA_1$) | 93.36 | 92.97 | 93.36 | 92.98 | 86.24 |
| RNX032 ($SA_1$) | 92.51 | 92.23 | 92.51 | 91.99 | 83.31 |
| RNX040 ($SA_1$) | 92.99 | 92.72 | 92.99 | 92.40 | 82.83 |
| RNX064 ($SA_1$) | 93.12 | 92.69 | 93.12 | 92.71 | 83.34 |
| RNX080 ($SA_1$) | 93.72 | 93.39 | 93.72 | 93.30 | 84.33 |
| RNX120 ($SA_1$) | 92.99 | 92.61 | 92.99 | 92.45 | 80.47 |
| RNX160 ($SA_1$) | 93.24 | 92.89 | 93.24 | 92.80 | 84.29 |
| RNX320 ($SA_1$) | 93.24 | 92.92 | 93.24 | 92.69 | 86.68 |
| RNX ($SA_2$) | 93.72 | 93.40 | 93.72 | 93.27 | 84.80 |
| RNY002 ($SA_1$) | 91.79 | 91.28 | 91.79 | 91.24 | 79.88 |
| RNY004 ($SA_1$) | 92.63 | 92.07 | 92.63 | 92.17 | 83.80 |
| RNY006 ($SA_1$) | 92.39 | 91.98 | 92.39 | 91.91 | 83.75 |
| RNY008 ($SA_1$) | 92.87 | 92.53 | 92.87 | 92.34 | 80.94 |
| RNY016 ($SA_1$) | 93.36 | 93.08 | 93.36 | 93.02 | 88.12 |
| RNY032 ($SA_1$) | 93.36 | 93.34 | 93.36 | 93.10 | 87.15 |
| RNY040 ($SA_1$) | 91.91 | 91.32 | 91.91 | 91.33 | 81.86 |
| RNY064 ($SA_1$) | 93.36 | 93.02 | 93.36 | 92.99 | 87.19 |
| RNY080 ($SA_1$) | 93.36 | 92.89 | 93.36 | 92.87 | 85.27 |
| RNY120 ($SA_1$) | 93.12 | 92.82 | 93.12 | 92.71 | 85.75 |
| RNY160 ($SA_1$) | 92.51 | 91.89 | 92.51 | 92.01 | 82.84 |
| RNY320 ($SA_1$) | 93.72 | 93.32 | 93.72 | 93.32 | 86.72 |
| RNY ($SA_2$) | 93.48 | 93.20 | 93.48 | 93.05 | 84.31 |
| RNXY ($SA_3$) | 93.48 | 93.16 | 93.48 | 93.02 | 84.32 |
| XY002 ($SA_2$) | 91.55 | 91.11 | 91.55 | 90.99 | 80.83 |
| XY004 ($SA_2$) | 92.75 | 92.30 | 92.75 | 92.20 | 82.35 |
| XY006 ($SA_2$) | 92.99 | 92.58 | 92.99 | 92.53 | 84.75 |
| XY008 ($SA_2$) | 93.72 | 93.48 | 93.72 | 93.33 | 84.80 |
| XY016 ($SA_2$) | 93.24 | 92.83 | 93.24 | 92.80 | 85.26 |
| XY032 ($SA_2$) | 93.48 | 93.31 | 93.48 | 93.07 | 85.27 |
| XY040 ($SA_2$) | 93.60 | 93.30 | 93.60 | 93.14 | 84.80 |
| XY064 ($SA_2$) | 93.24 | 92.81 | 93.24 | 92.84 | 84.79 |
| XY080 ($SA_2$) | 93.84 | 93.56 | 93.84 | 93.48 | 84.33 |
| XY120 ($SA_2$) | 93.12 | 92.74 | 93.12 | 92.67 | 82.86 |
| XY160 ($SA_2$) | 92.99 | 92.53 | 92.99 | 92.43 | 82.85 |
| XY320 ($SA_2$) | 93.96 | 93.60 | 93.96 | 93.60 | 87.22 |
| RN_XY ($SA_3$) | 93.84 | 93.56 | 93.84 | 93.48 | 84.33 |
| RN ($SA_4$) | 93.48 | 93.16 | 93.48 | 93.02 | 84.32 |
| **Ours** ($CWE_4$) | 94.08 | 93.77 | 94.08 | 93.71 | 85.78 |

For example, at $MV_1$, models like RNX008 and RNX064 show reasonably strong performance with accuracies of 92.75% and 93.36%, respectively. Yet, this upward trend is not consistently maintained across all layers. The final predictions at $MV_4$ for RN achieve an accuracy of 93.84%, which, while solid, fails to outperform the best results obtained with our proposed CWE method.

Notably, our $CWE_4$ model surpasses MV's performance with an accuracy of 94.08%, F1-score of 93.71%, and all other metrics. These results underscore the effectiveness of our CWE approach, which optimally combines classifier outputs, outperforming traditional ensemble methods like MV, particularly at higher layers where performance is expected to

**Table 22. Performance evaluation using MV instead of CWE.**

| Algorithm | Accuracy | Precision | Recall | F1-score | Specificity |
|---|---|---|---|---|---|
| RNX002 ($MV_1$) | 91.55 | 90.93 | 91.55 | 91.11 | 83.70 |
| RNX004 ($MV_1$) | 92.87 | 92.41 | 92.87 | 92.56 | 86.20 |
| RNX006 ($MV_1$) | 92.39 | 91.93 | 92.39 | 91.96 | 84.72 |
| RNX008 ($MV_1$) | 92.75 | 92.42 | 92.75 | 92.33 | 82.38 |
| RNX016 ($MV_1$) | 92.75 | 92.60 | 92.75 | 92.55 | 89.52 |
| RNX032 ($MV_1$) | 92.63 | 92.89 | 92.63 | 92.53 | 89.47 |
| RNX040 ($MV_1$) | 92.15 | 91.77 | 92.15 | 91.78 | 84.74 |
| RNX064 ($MV_1$) | 93.36 | 92.99 | 93.36 | 93.09 | 88.62 |
| RNX080 ($MV_1$) | 93.60 | 93.33 | 93.60 | 93.26 | 88.16 |
| RNX120 ($MV_1$) | 91.55 | 91.02 | 91.55 | 91.03 | 85.16 |
| RNX160 ($MV_1$) | 92.15 | 91.41 | 92.15 | 91.60 | 82.83 |
| RNX320 ($MV_1$) | 93.36 | 93.04 | 93.36 | 92.93 | 85.29 |
| RNX ($MV_2$) | 93.60 | 93.43 | 93.60 | 93.10 | 83.37 |
| RNY002 ($MV_1$) | 91.06 | 90.24 | 91.06 | 90.39 | 81.79 |
| RNY004 ($MV_1$) | 92.51 | 92.03 | 92.51 | 91.98 | 82.83 |
| RNY006 ($MV_1$) | 92.03 | 91.59 | 92.03 | 91.40 | 80.41 |
| RNY008 ($MV_1$) | 92.39 | 91.94 | 92.39 | 92.03 | 84.27 |
| RNY016 ($MV_1$) | 92.63 | 92.13 | 92.63 | 92.08 | 85.71 |
| RNY032 ($MV_1$) | 93.48 | 93.16 | 93.48 | 93.01 | 85.28 |
| RNY040 ($MV_1$) | 92.75 | 92.58 | 92.75 | 92.40 | 86.19 |
| RNY064 ($MV_1$) | 92.75 | 92.29 | 92.75 | 92.40 | 84.29 |
| RNY080 ($MV_1$) | 93.60 | 93.33 | 93.60 | 93.26 | 88.16 |
| RNY120 ($MV_1$) | 93.72 | 93.59 | 93.72 | 93.43 | 88.62 |
| RNY160 ($MV_1$) | 92.75 | 92.40 | 92.75 | 92.54 | 87.64 |
| RNY320 ($MV_1$) | 93.36 | 93.04 | 93.36 | 92.93 | 85.29 |
| RNY ($MV_2$) | 93.72 | 93.39 | 93.72 | 93.26 | 83.85 |
| RNXY ($MV_3$) | 93.84 | 93.54 | 93.84 | 93.42 | 83.86 |
| XY002 ($MV_2$) | 92.27 | 92.20 | 92.27 | 91.96 | 88.52 |
| XY004 ($MV_2$) | 92.75 | 92.59 | 92.75 | 92.63 | 90.50 |
| XY006 ($MV_2$) | 92.99 | 93.00 | 92.99 | 92.83 | 89.98 |
| XY008 ($MV_2$) | 92.99 | 92.90 | 92.99 | 92.92 | 89.10 |
| XY016 ($MV_2$) | 92.87 | 92.96 | 92.87 | 92.83 | 91.46 |
| XY032 ($MV_2$) | 91.43 | 91.84 | 91.43 | 91.47 | 90.38 |
| XY040 ($MV_2$) | 92.75 | 92.58 | 92.75 | 92.57 | 89.56 |
| XY064 ($MV_2$) | 93.72 | 93.80 | 93.72 | 93.73 | 92.94 |
| XY080 ($MV_2$) | 93.84 | 93.74 | 93.84 | 93.72 | 91.04 |
| XY120 ($MV_2$) | 93.96 | 93.85 | 93.96 | 93.79 | 90.57 |
| XY160 ($MV_2$) | 92.87 | 92.62 | 92.87 | 92.70 | 90.02 |
| XY320 ($MV_2$) | 92.39 | 92.49 | 92.39 | 92.20 | 91.44 |
| RN_XY ($MV_3$) | 93.72 | 93.39 | 93.72 | 93.26 | 83.85 |
| RN ($MV_4$) | 93.84 | 93.69 | 93.84 | 93.67 | 90.55 |
| **Ours ( $CWE_4$)** | 94.08 | 93.77 | 94.08 | 93.71 | 85.78 |

peak. The consistency and superiority of $CWE_4$ highlight its robustness and advantage over MV, even when the latter is executed perfectly. Thus, it is clear that the highest accuracy achieved by $MV_4$ is 0.24% lower than that of our $CWE_4$.

*4.5.2.3 Weighted Averaging (WA):*

As depicted in the next table, Weighted Averaging (WA) with all predictions at Layer $L$ is denoted as $WA_L$. The performance of WA across various layers is presented in Table 23, where random weights are assigned to each classifier, ensuring that the sum of all weights equals one. The weights are allocated such that the highest-performing models receive the greatest weights, followed by progressively lower weights for less effective models.

**Table 23. Performance evaluation using WA instead of CWE.**

| Algorithm | Accuracy | Precision | Recall | F1-score | Specificity |
|---|---|---|---|---|---|
| RNX002 ($WA_1$) | 92.27 | 91.90 | 92.27 | 91.84 | 84.24 |
| RNX004 ($WA_1$) | 92.87 | 92.54 | 92.87 | 92.48 | 85.22 |
| RNX006 ($WA_1$) | 92.39 | 92.07 | 92.39 | 91.89 | 83.74 |
| RNX008 ($WA_1$) | 93.36 | 93.08 | 93.36 | 93.04 | 85.74 |
| RNX016 ($WA_1$) | 93.24 | 92.83 | 93.24 | 92.84 | 85.76 |
| RNX032 ($WA_1$) | 92.87 | 92.67 | 92.87 | 92.39 | 83.81 |
| RNX040 ($WA_1$) | 92.99 | 92.62 | 92.99 | 92.42 | 83.31 |
| RNX064 ($WA_1$) | 92.75 | 92.26 | 92.75 | 92.32 | 83.32 |
| RNX080 ($WA_1$) | 93.96 | 93.65 | 93.96 | 93.59 | 84.83 |
| RNX120 ($WA_1$) | 93.36 | 93.06 | 93.36 | 92.83 | 80.49 |
| RNX160 ($WA_1$) | 92.87 | 92.45 | 92.87 | 92.41 | 84.75 |
| RNX320 ($WA_1$) | 93.36 | 93.04 | 93.36 | 92.91 | 87.64 |
| RNX ($WA_2$) | 93.72 | 93.40 | 93.72 | 93.27 | 84.80 |
| RNY002 ($WA_1$) | 91.30 | 90.75 | 91.30 | 90.73 | 79.37 |
| RNY004 ($WA_1$) | 92.75 | 92.20 | 92.75 | 92.31 | 84.29 |
| RNY006 ($WA_1$) | 92.75 | 92.42 | 92.75 | 92.36 | 84.72 |
| RNY008 ($WA_1$) | 92.87 | 92.55 | 92.87 | 92.36 | 81.41 |
| RNY016 ($WA_1$) | 93.48 | 93.27 | 93.48 | 93.14 | 88.60 |
| RNY032 ($WA_1$) | 93.72 | 93.67 | 93.73 | 93.47 | 87.18 |
| RNY040 ($WA_1$) | 92.51 | 92.14 | 92.51 | 91.94 | 82.36 |
| RNY064 ($WA_1$) | 93.36 | 93.02 | 93.36 | 93.05 | 88.15 |
| RNY080 ($WA_1$) | 93.36 | 92.98 | 93.36 | 92.89 | 85.27 |
| RNY120 ($WA_1$) | 93.36 | 93.12 | 93.36 | 93.06 | 87.18 |
| RNY160 ($WA_1$) | 92.63 | 92.02 | 92.63 | 92.15 | 83.33 |
| RNY320 ($WA_1$) | 93.72 | 93.33 | 93.72 | 93.35 | 86.72 |
| RNY ($WA_2$) | 93.96 | 93.72 | 93.96 | 93.55 | 85.78 |
| RNXY ($WA_3$) | 93.60 | 93.29 | 93.60 | 93.15 | 84.80 |
| XY002 ($WA_2$) | 91.55 | 91.03 | 91.55 | 91.05 | 79.39 |
| XY004 ($WA_2$) | 93.12 | 92.66 | 93.12 | 92.65 | 83.32 |
| XY006 ($WA_2$) | 92.87 | 92.47 | 92.87 | 92.41 | 84.75 |
| XY008 ($WA_2$) | 93.72 | 93.46 | 93.72 | 93.37 | 85.28 |
| XY016 ($WA_2$) | 93.12 | 92.72 | 93.12 | 92.69 | 85.25 |
| XY032 ($WA_2$) | 93.60 | 93.44 | 93.60 | 93.23 | 85.75 |
| XY040 ($WA_2$) | 93.48 | 93.19 | 93.48 | 93.04 | 84.32 |
| XY064 ($WA_2$) | 93.36 | 92.92 | 93.36 | 92.96 | 84.80 |
| XY080 ($WA_2$) | 93.96 | 93.69 | 93.96 | 93.60 | 84.35 |
| XY120 ($WA_2$) | 93.48 | 93.16 | 93.48 | 93.10 | 83.35 |
| XY160 ($WA_2$) | 92.87 | 92.42 | 92.87 | 92.31 | 82.85 |
| XY320 ($WA_2$) | 93.60 | 93.20 | 93.60 | 93.20 | 85.76 |
| RN_XY ($WA_3$) | 93.72 | 93.41 | 93.72 | 93.27 | 84.81 |
| RN ($WA_4$) | 93.60 | 93.29 | 93.60 | 93.15 | 84.80 |
| **Ours** ($CWE_4$) | 94.08 | 93.77 | 94.08 | 93.71 | 85.78 |

The results show that while some configurations exhibit strong performance, particularly with higher accuracy and F1-scores in the upper layers, the final results ($WA_4$) still fall short when compared to our proposed method. For instance, at $WA_1$, accuracy values are notably high for several models, but the performance does not consistently improve across the layers. Specifically, the final layer $WA_4$ achieves an accuracy of 93.60% and an F1-score of 93.15%, which, despite being competitive, does not surpass the results obtained with our CWE method.

In conclusion, although Weighted Averaging demonstrates reasonable performance across different layers and configurations, it does not achieve the same level of effectiveness as our

CWE model. The CWE approach outperforms WA, highlighting its superior capability in leveraging classifier predictions for enhanced overall performance. So, it is evident from our findings that $WA_4$ achieves an accuracy that is 0.48% lower than our $CWE_4$.

Across all tables and metrics, CWE demonstrates superior performance compared to any other conventional ensemble method. The consistent improvement in accuracy, precision, recall, F1-score, and specificity across different classifier ensembles proves that CWE is a more effective method for combining classifier outputs, resulting in better overall model performance.

**4.5.3 ORNS architectures in single-layer CWE**   In the single-layer CWE approach, we establish evaluation metrics to reassess the claim regarding the superiority of the multi-layer CWE. The results, as shown in Table 24, clearly indicate that the multi-layer CWE significantly outperforms the single-layer version, which achieves only 93.48% accuracy. This difference is further supported by the values of other performance metrics, confirming the robustness of the claim.

The comprehensive performance comparisons clearly demonstrate that our approach, which integrates AT with CWE across multiple layers, offers a superior architecture compared to existing methods.

## 4.6 Answers to the research questions

*Answer to RQ1:* To optimize the Transfer Learning model for our classification tasks, we strategically employ all variants of RegNet. By harnessing RegNet's diverse architectural patterns, we explore a wide design space, ensuring our models are well-suited for various tasks. Fine-tuning with customized layers allows us to tailor each model to the specific requirements of each task. This approach leverages RegNet's scalable and efficient quantized linear parameterization, enhancing performance while maintaining computational efficiency. Ultimately, combining RegNet's robust architecture with targeted fine-tuning results in a highly effective TL model, demonstrating RegNet's versatility in TL applications.

*Answer to RQ2:* To effectively identify the most critical features, particularly significant areas or regions, the Attention-Triplet (AT) approach is applied in the design of an ORNS model. This method prioritizes important features by focusing more on them as feature maps pass through different layers, enhancing the model's ability to capture relevant details. It balances the risks associated with ignoring deep features in simpler models or overfitting in more complex ones. By incorporating AT into the ORNS architecture, the model demonstrates superior performance compared to versions without an attention mechanism.

*Answer to RQ3:* A single algorithm is not sufficient for skin lesion classification; instead, an Ensemble Learning (EL) technique is necessary. Our study shows that using multiple classifiers in an ensemble is more effective than relying on a single classifier. By integrating a customized CNN architecture with three different attention mechanisms (CA, SEA, and SA) and combining them with TL models, we construct various ORNS architectures. Testing these architectures across 24 models, including all RegNet variants, and applying diverse ensemble strategies for predictions reveal a significant improvement in handling unseen data.

Table 24. Performance evaluation of single-layer CWE.

| Algorithm | Accuracy | Precision | Recall | F1-score | Specificity |
|---|---|---|---|---|---|
| *SL−CWE* | 93.48 | 93.16 | 93.48 | 93.02 | 84.32 |
| **Ours** | 94.08 | 93.77 | 94.08 | 93.71 | 85.78 |

This approach enhances both the accuracy and robustness of skin lesion classification by leveraging diverse methods for feature extraction and classification.

*Answer to RQ4:* Traditional Ensemble Learning (EL) methods have limitations in effectively weighting each model's contribution, often relying on fixed or manually determined weights, which can lead to suboptimal performance on unseen data. The proposed approach overcomes these limitations by dynamically calculating optimal weight ratios for each model using a novel method based on $\chi^2$ values. This data-driven technique allows for better generalization and performance by selecting the optimal ratio of predictions from each model. Visual results from the ML-CWE demonstrate significant performance improvements, highlighting increased robustness and efficiency in managing diverse patterns for complex DL tasks.

## 5 Discussion and extended comparison

Our study demonstrates the effectiveness of combining ORNS architectures with a sophisticated ensemble approach, CWE, for enhancing classification performance. By integrating data preprocessing, augmentation, and fine-tuning of RegNet models, we address class imbalances and optimize feature extraction, leading to significant improvements in model accuracy.

The final results, achieved at CWE-Layer 4, show that our approach effectively leverages iterative refinement and model ensembling. The RN model, with an accuracy of 94.08%, outperforms its predecessors, highlighting the success of our multi-layer CWE strategy. The high specificity of 85.78% further demonstrates the model's capability in distinguishing non-target classes, ensuring reliable performance across various metrics.

These outcomes validate our methodological choices, including the application of advanced attention mechanisms and ensemble techniques. The enhanced performance metrics underscore the potential of our approach to achieve robust and accurate predictions, making it a valuable contribution to the field of image classification.

Despite a larger array of classifiers, our advanced yet user-friendly approach demonstrates clear superiority in overall performance, generalization, and evaluation measures in this domain. A comprehensive comparison of our proposed model's performance against existing literature is detailed in Table 25. Notably, we let alone compare the studies conducted using the HAM10000 dataset.

Additionally, we compared the performance of our architecture with state-of-the-art methods to demonstrate its superiority. As evidenced in Table 26, our customized architecture outperforms existing approaches, successfully validating its effectiveness.

### 5.1 Advantages and disadvantages of CWE

The CWE method offers several advantages and a few limitations compared to other weighting schemes. Its primary advantage lies in its statistical foundation, leveraging the $Chi^2$ statistic to assign weights based on classifier performance. This ensures that models demonstrating a stronger relationship between predictions and actual outcomes have a greater influence, thereby enhancing the reliability and accuracy of the ensemble. Additionally, the CWE method is adaptable and can be applied in multi-layer scenarios, allowing the incremental emphasis on high-performing classifiers. Finally, guaranteeing the optimal weight for each classification model can be as the most important advantage.

However, CWE relies heavily on the quality of the $Chi^2$ test results, making it sensitive to small sample sizes or imbalanced datasets, which could skew the weight assignments. Moreover, its computational complexity may increase when dealing with large-scale datasets or a high number of classifiers due to the need for calculating and normalizing values. Despite

**Table 25. Comparison of our proposed architecture with existing others.**

| Article | Accuracy | Precision | Recall | F1-Score | Specificity |
|---------|----------|-----------|--------|----------|-------------|
| [7] | 84.30 | - | - | - | - |
| [11] | 86.67 | - | - | - | - |
| [12] | 80.00 | - | - | - | - |
| [13] | 91.51 | - | - | - | - |
| [14] | 94.00 | - | - | - | - |
| [15] | 86.33 | - | 86.33 | - | 97.48 |
| [16] | 89.50 | - | 89.50 | - | **98.10** |
| [17] | 91.51 | - | - | - | - |
| [20] | 91.24 | 83.53 | 95.04 | 88.91 | - |
| [21] | 86.20 | 91.30 | - | - | - |
| [22] | 93.46 | - | - | - | 92.90 |
| [23] | 88.00 | 87.00 | 94.00 | 89.00 | - |
| [24] | 86.20 | - | - | - | - |
| [25] | 93.45 | 93.57 | 93.01 | 93.45 | - |
| [26] | 83.20 | - | - | - | - |
| [27] | 93.46 | 87.01 | 85.57 | 86.28 | - |
| [28] | 93.40 | 93.7 | - | - | - |
| [29] | 90.00 | 86.00 | 81.00 | 86.00 | - |
| **Ours** | **94.08** | **93.77** | **94.08** | **93.71** | 85.78 |

**Table 26. Comparison of our proposed architecture with state-of-the-art methods.**

| Model | Acc | Pre | Re | F1 | Spe |
|-------|-----|-----|-----|-----|-----|
| EfficientNetB0 | 92.03 | 91.26 | 92.03 | 91.45 | 81.87 |
| EfficientNetB1 | 92.63 | 92.45 | 92.63 | 92.35 | 85.69 |
| EfficientNetB2 | 92.51 | 91.95 | 92.51 | 92.05 | 84.28 |
| EfficientNetB3 | 92.63 | 92.51 | 92.63 | 92.41 | 86.70 |
| EfficientNetB4 | 92.15 | 91.76 | 92.15 | 91.73 | 83.29 |
| EfficientNetB5 | 92.15 | 92.41 | 92.15 | 92.16 | 90.47 |
| EfficientNetB6 | 91.79 | 91.40 | 91.79 | 91.44 | 84.69 |
| EfficientNetB7 | 92.51 | 91.93 | 92.51 | 92.04 | **86.65** |
| ResNet50 | 91.91 | 91.51 | 91.91 | 91.47 | 84.23 |
| ResNet101 | 91.18 | 90.81 | 91.18 | 90.78 | 82.26 |
| ResNet152 | 91.30 | 91.41 | 91.30 | 91.18 | 85.61 |
| DenseNet121 | 92.39 | 91.99 | 92.39 | 92.12 | **86.65** |
| DenseNet169 | 92.75 | 92.54 | 92.75 | 92.39 | 85.75 |
| DenseNet201 | 91.18 | 90.92 | 91.18 | 90.94 | 86.15 |
| MobileNet | 91.42 | 90.90 | 91.42 | 90.86 | 80.86 |
| MobileNetV2 | 91.90 | 91.80 | 91.90 | 91.60 | 83.31 |
| MobileNetV3 Large | 92.99 | 92.60 | 92.99 | 92.34 | 85.25 |
| InceptionV3 | 92.27 | 91.86 | 92.27 | 91.62 | 85.19 |
| InceptionResNetV2 | 90.70 | 89.85 | 90.70 | 90.01 | 79.39 |
| Xception | 90.45 | 89.96 | 90.45 | 90.15 | 84.18 |
| **ML-CWE** | **94.08** | **93.77** | **94.08** | **93.71** | 85.78 |

these limitations, CWE is particularly advantageous for scenarios requiring robust and statistically sound ensembling strategies.

## 5.2 Threats to validity

Although our approach shows vast significance in image classification, outlined below are certain limitations of our study, which may serve as opportunities for future research and refinement:

*Extensive Use of Classifiers in the Initial Ensemble Layer*

Incorporating a diverse set of classifiers in the initial layer can enhance model robustness, but it also introduces significant computational demands. The increased complexity and resource requirements during both training and inference limit the scalability of our approach, particularly when applied to larger datasets or in environments with limited computing resources.

*Dependence on a Single Dataset*

Our study's reliance on a single dataset for training and evaluation poses a potential limitation in terms of generalizability. A model trained on just one dataset may not capture the full range of variations across different data sources, potentially leading to reduced performance when applied to new and unseen datasets.

# 6 Future work and research directions

Our future work will address several key research gaps and explore areas that require further investigation to enhance the applicability and efficiency of our model.

*Streamlined Web-Based API for Real-Time Diagnostic Predictions:*

We plan to develop a more user-friendly, web-based API that allows for real-time diagnostic predictions. This would provide dermatologists and patients with a practical tool for on-demand skin lesion analysis. The API will improve accessibility and scalability, allowing users to submit skin images from anywhere and receive immediate feedback on potential skin conditions.

*Evaluation Across Multiple Datasets:*

To ensure our model's robustness and generalizability, we intend to test its performance on a variety of diverse datasets. This will help assess the model's ability to handle different types of skin lesions, lighting conditions, and image quality. Such an evaluation will provide valuable insights into how the model performs across real-world scenarios and across different populations.

*Reduction of Computational Costs:*

While our model achieves high accuracy, computational efficiency remains a key area for improvement. In future research, we will focus on optimizing the architecture by reducing the number of algorithms used, while maintaining or even improving accuracy. This will help reduce computational costs, making the model more feasible for deployment in resource-constrained environments, such as mobile devices or remote healthcare settings.

These research directions aim to address the challenges of scalability, accessibility, and efficiency, ensuring that our model can be widely adopted and deployed to improve early diagnosis and patient care in dermatology.

# 7 Conclusion

This paper presents a comprehensive approach to image classification through the innovative application of ORNS architectures and a Multi-Layer $\chi^2$ Weighted Ensemble (ML-CWE) strategy. Our methodology begins with meticulous data preprocessing and augmentation, leading to the effective training and fine-tuning of diverse ORNS models. The integration of three advanced attention mechanisms enhances the model's ability to focus on critical features and improve performance.

The Multi-Layer CWE technique, comprising sequential refinement across four layers, proves to be a powerful tool for optimizing ensemble predictions. By leveraging the strengths of various models at each layer, the final RN model achieves remarkable performance. This iterative approach effectively harnesses the benefits of each model, culminating in a robust and high-performing classification system.

Our study highlights the potential of ORNS architectures and Multi-Layer CWE to advance image classification through improved accuracy, reliability, and interpretability, as demonstrated by GradCAM visualizations. These contributions set new benchmarks in Transfer Learning and hold significant practical value, particularly in healthcare. By enabling early diagnosis of skin conditions, the approach can improve patient outcomes and accessibility.

All data used in this study, including the augmented training data, are publicly accessible on the Kaggle repository: [HAM10000 Dataset] [31] (https://www.kaggle.com/datasets/anwar hossaine/ham10000-splitted-and-augmented-CWE-70-15-15). Our use of the HAM10000 dataset complies with the Creative Commons Attribution-NonCommercial 4.0 International Public License because we have properly attributed the original dataset and cited the recommended paper by the authors. This fulfills the attribution requirement of the license. Additionally, our use of the dataset is for non-commercial purposes, aligning with the non-commercial clause of the license.

## Author contributions

**Conceptualization:** Anwar Hossain Efat.

**Data curation:** Anwar Hossain Efat.

**Formal analysis:** Anwar Hossain Efat.

**Investigation:** Anwar Hossain Efat.

**Methodology:** Anwar Hossain Efat.

**Project administration:** Anwar Hossain Efat.

**Software:** Anwar Hossain Efat.

**Validation:** Anwar Hossain Efat.

**Visualization:** Anwar Hossain Efat.

**Writing – original draft:** Anwar Hossain Efat.

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
