## [Decision Letter · Decision Letter 0]

17 Dec 2024

PONE-D-24-37946Chi2 Weighted Ensemble: A Multi-Layer Ensemble Approach for Skin Lesion Classification using A Novel Framework - Optimized RegNet Synergy with Attention-TripletPLOS ONE

Dear Dr. Efat,

Thank you for submitting your manuscript to PLOS ONE. After careful consideration, we feel that it has merit but does not fully meet PLOS ONE’s publication criteria as it currently stands. Therefore, we invite you to submit a revised version of the manuscript that addresses the points raised during the review process.

We look forward to receiving your revised manuscript.

Kind regards,

Veer Singh, Ph.D

Academic Editor

PLOS ONE

Journal Requirements:

Reviewers' comments:

Reviewer's Responses to Questions

**Comments to the Author**

1. Is the manuscript technically sound, and do the data support the conclusions?

Reviewer #1: No

Reviewer #2: Yes

Reviewer #3: Partly

2. Has the statistical analysis been performed appropriately and rigorously? 

Reviewer #1: No

Reviewer #2: Yes

Reviewer #3: Yes

3. Have the authors made all data underlying the findings in their manuscript fully available?

Reviewer #1: No

Reviewer #2: Yes

Reviewer #3: Yes

4. Is the manuscript presented in an intelligible fashion and written in standard English?

Reviewer #1: No

Reviewer #2: No

Reviewer #3: No

5. Review Comments to the Author

Reviewer #1: The manuscript fails to meet the necessary standards for publication due to several significant issues. Firstly, the proposed Transfer Learning-based framework lacks sufficient novelty. Many of the techniques mentioned, such as the use of RegNet architectures, attention mechanisms, and Ensemble Learning, are already well-established in the field. While the authors introduce the $Chi^2$ Weighted Ensemble method, the manuscript provides insufficient detail on how this method differs meaningfully from existing ensemble techniques. Moreover, the Multi-Layer $Chi^2$ Weighted Ensemble appears to be an incremental improvement rather than a breakthrough innovation, and the manuscript lacks clarity on how this method substantively enhances model performance over traditional approaches.

Additionally, while the manuscript reports high accuracy on the HAM1000 dataset, there is a lack of comprehensive evaluation. The authors rely heavily on a single dataset, which raises concerns about the generalizability and robustness of the proposed approach. The manuscript does not discuss performance on diverse or real-world datasets, which is essential for validating the broad applicability of any new technique, especially in critical applications like skin lesion detection.

Furthermore, the interpretability aspect, while mentioned through the use of Gradient Class Activation Maps, is not explored in depth. The manuscript fails to provide sufficient evidence that the proposed method significantly enhances transparency or clinical relevance compared to existing interpretability techniques. The claims regarding improvements in early diagnosis, time, accessibility, and cost reduction are also not adequately supported by empirical evidence or detailed analysis.

Finally, the manuscript suffers from a lack of clarity in explaining how the integration of the various attention mechanisms and the Ensemble Learning strategy leads to superior results. The methodology section lacks a clear, cohesive explanation, making it difficult to follow the progression of the approach from inception to evaluation. Overall, the manuscript does not offer enough innovation, depth of evaluation, or clarity in presentation to warrant publication.

Reviewer #2: The manuscript titled " Chi2 Weighted Ensemble: A Multi-Layer Ensemble Approach for Skin Lesion Classification using A Novel Framework - Optimized RegNet Synergy with Attention-Triplet” is found to be well-structured. However, the manuscript needs to improve significantly. However, I have suggested revision that I believe will further enhance the clarity and impact of your manuscript:

The abstract section needs to be updated.

I suggest highlighting specific contributions or insights gained from the research in the abstract section. This will help differentiate your study from existing literature reviews on the topic and emphasize its novelty.

The quality of the figures needs to be improved in the revised manuscript.

Arrange significant figures throughout the manuscript.

I recommend incorporating the recent papers that have been discussed and cited in the manuscript.

Manuscript briefly elaborated on specific research gaps or areas that require further investigation. This could help readers understand the potential directions for future studies. Please add a future direction section or include it with the conclusion.

Please make sure to define each acronym at its first use. Check through the entire manuscript to make sure it is defined at the first use.

The English of the manuscript needs to be improved.

The conclusion section should be revised and concise.

Reviewer #3: Dr. Anwar Hossain Efat's manuscript describes the Chi2 Weighted Ensemble: A multi-layer ensemble approach for skin Lesion classification using A novel framework- optimized regnet synergy with attention-triplet. This approach has been reported in the literature several times, and regular studies have been conducted on Transfer Learning-based frameworks incorporating Optimized RegNet Synergy architectures and Attention-Triplet, including channel attention, squeeze-excitation attention, and soft attention, combined with an advanced Ensemble Learning Strategy.

What distinguishes the work of Dr. Anwar Hossain Efat is that it contributes to the early diagnosis of skin conditions and reduces the risks associated with neglect. However, the present protocol is very attractive from the viewpoint of this journal.

So, I recommend publication in this journal; However, the manuscript needs major revisions before publication, particularly regarding the scope of the process.

Some suggestions are listed below:

1. The alignment of the manuscript looks so poor (justify all the paragraphs).

2. Introduction requires more references to validate the author's points.

3. In terms of novelty, what are the potential practical applications of this research, particularly in the early detection and diagnosis of skin cancer?

4. How were the hyperparameters of the Optimized RegNet Synergy architectures and the Attention-Triplet modules tuned?

5. The study reports an impressive accuracy of 94.08% on the HAM1000 dataset. How does this performance compare to other state-of-the-art methods, and what factors contribute to the improved accuracy? (please include in the manuscript.)

6. The use of Gradient Class Activation Maps (Grad-CAM) is a valuable tool for model interpretability. Could you provide more insights into the specific regions highlighted by Grad-CAM and their relevance to the classification decisions?

7. How does the proposed Multi-Layer Chi2 Weighted Ensemble differ from other ensemble methods, such as bagging, boosting, and stacking?

8. Include the advantages and disadvantages of the Chi2 Weighted Ensemble method compared to other weighting schemes.

9. Figures 5 and 6,8 should be redrawn for the general readership of the audience.

6. PLOS authors have the option to publish the peer review history of their article (what does this mean?). If published, this will include your full peer review and any attached files.

Reviewer #1: **Yes: **Sachchida Nand Rai

Reviewer #2: No

Reviewer #3: **Yes: **Prasanth Thumpati

---

## [Author Response · Author response to Decision Letter 1]

25 Dec 2024

To the Editor

Comment# 1: Please ensure that your manuscript meets PLOS ONE's style requirements, including those for file naming. The PLOS ONE style templates can be found at

Author response: Thanks for reminding us to follow the journal formatting requirements.

Author action: We have written the entire paper in Latex using PLOS ONE’s suggested template collected from the cite: https://journals.plos.org/plosone/s/latex. Additionally, we have also rechecked the format after generation of the pdf. As a result, we can confirm that, the requirement is correctly satisfied.

Comment# 2: Please note that PLOS ONE has specific guidelines on code sharing for submissions in which author-generated code underpins the findings in the manuscript. In these cases, all author-generated code must be made available without restrictions upon publication of the work. Please review our guidelines at https://journals.plos.org/plosone/s/materials-and-software-sharing#loc-sharing-code and ensure that your code is shared in a way that follows best practice and facilitates reproducibility and reuse.

Author response: Thanks for the comment.

Author action: Regarding the availability of the code, the authors assert that, due to ethical considerations, the code cannot be made publicly available. However, readers requiring access for manuscript assessment purposes may request the code. To do so, a formal request should be sent to the corresponding author.

To Reviewer 1

Reviewer#1, Comment # 1: The manuscript fails to meet the necessary standards for publication due to several significant issues. Firstly, the proposed Transfer Learning-based framework lacks sufficient novelty. Many of the techniques mentioned, such as the use of RegNet architectures, attention mechanisms, and Ensemble Learning, are already well-established in the field. While the authors introduce the $Chi^2$ Weighted Ensemble method, the manuscript provides insufficient detail on how this method differs meaningfully from existing ensemble techniques. Moreover, the Multi-Layer $Chi^2$ Weighted Ensemble appears to be an incremental improvement rather than a breakthrough innovation, and the manuscript lacks clarity on how this method substantively enhances model performance over traditional approaches.

Author response: We sincerely thank the reviewer for this valuable concern.

Author action: We would like to clarify that our proposed Transfer Learning-based framework has significant novelty. While techniques like RegNet architectures, attention mechanisms, and Ensemble Learning are well-established, our approach is unique in how we integrate three distinct attention modules—channel attention, squeeze-excitation attention, and soft attention—within a single architecture to fine-tune pre-trained models. This is a novel application that has not been explored in previous studies.

We acknowledge that RegNet architectures and attention mechanisms are established, but no previous study has integrated all these components into a single framework as we have done. Furthermore, the $Chi^2$ Weighted Ensemble (CWE) method we introduce is not a conventional ensemble technique; it is an entirely new approach to weighting model predictions, and this innovation differentiates it from existing methods.

Our manuscript provides detailed explanations of how CWE differs from traditional ensemble techniques, with dedicated sections (Sections 3.6 and 3.7) outlining its unique characteristics. Additionally, Figures 7 and 8 further illustrate the working principles of CWE.

Regarding the performance, we have conducted an ablation study in Section 4.5, which clearly demonstrates that our novel approach contributes significantly to improving model performance. The results from this study validate the added value of our methodology in this domain.

Reviewer#1, Comment # 2: Additionally, while the manuscript reports high accuracy on the HAM1000 dataset, there is a lack of comprehensive evaluation. The authors rely heavily on a single dataset, which raises concerns about the generalizability and robustness of the proposed approach. The manuscript does not discuss performance on diverse or real-world datasets, which is essential for validating the broad applicability of any new technique, especially in critical applications like skin lesion detection.

Author response: We greatly appreciate the reviewer’s insightful concern about using single dataset.

Author action: We would like to clarify that we have provided a comprehensive evaluation of our proposed approach. In addition to reporting accuracy, we have also presented precision, recall, F1 score, and specificity to give a more thorough assessment of the model's performance. Furthermore, we included the ROC-AUC curve to demonstrate that the training process was not overfitted.

Our architecture, being a novel combination of various versions of RegNet, required extensive evaluation, which we believe provides a robust understanding of its performance. While we appreciate the importance of using multiple datasets for generalization, we have included this as part of our future direction.

The HAM1000 dataset we used, is a real-world dataset, and we took extra care to evaluate it in a manner that mimics real-world scenarios. Specifically, we reserved a separate test set that was not used in any preprocessing steps to ensure the data's independence and the generalizability of the results.

Reviewer#1, Comment # 3: Furthermore, the interpretability aspect, while mentioned through the use of Gradient Class Activation Maps, is not explored in depth. The manuscript fails to provide sufficient evidence that the proposed method significantly enhances transparency or clinical relevance compared to existing interpretability techniques. The claims regarding improvements in early diagnosis, time, accessibility, and cost reduction are also not adequately supported by empirical evidence or detailed analysis.

Author response: We sincerely appreciate the reviewer’s significant comment.

Author action: We appreciate your concerns regarding the interpretability aspect of our proposed method. To address this, we have expanded the discussion of Gradient Class Activation Maps (Grad-CAM) in the manuscript including the implementation process in Figure 12 in page 30. Specifically, we have elaborated on how Grad-CAM highlights key regions of interest within skin lesions, thereby enhancing model transparency and interpretability. By visually demonstrating which areas of the image are most influential in the model’s decision-making process, we have shown that our method significantly improves interpretability compared to conventional techniques. The enhanced transparency not only supports clinical decision-making but also increases trust in the model’s predictions.

Our GradCAM visualization provides significant details in Figure 13 that how a model emphasizes on each class images. Moreover, how the attention modules indicate the key region, also included in our manuscript in Figure 14.

Regarding the claims about early diagnosis, time, accessibility, and cost reduction, we agree that empirical evidence would further strengthen these claims. We admit that we haven’t checked it in real world but hypothetically we have shown the importance of our early detection architecture and there needs no cost for it. Additionally, the cost-effectiveness of the proposed model, particularly when deployed via a web-based API for real-time predictions, could alleviate the need for expensive hospital visits for initial screening, which would reduce healthcare costs. We plan to explore these aspects further in future work by collecting user feedback and conducting pilot studies to provide concrete evidence of the real-world impact of our approach.

Reviewer#1, Comment # 4: Finally, the manuscript suffers from a lack of clarity in explaining how the integration of the various attention mechanisms and the Ensemble Learning strategy leads to superior results. The methodology section lacks a clear, cohesive explanation, making it difficult to follow the progression of the approach from inception to evaluation. Overall, the manuscript does not offer enough innovation, depth of evaluation, or clarity in presentation to warrant publication.

Author response: We sincerely appreciate the reviewer’s valuable feedback.

Author action: We would like to clarify that the manuscript already provides a detailed explanation of the integration of attention mechanisms and the Ensemble Learning strategy.

1. Integration of Attention Mechanisms: The methodology section explicitly outlines how the channel attention, squeeze-excitation attention, and soft attention modules were integrated within the proposed architecture. Figure 5 properly depicts the way of integration of attention modules. Moreover, the dedicated ‘Section 3.5 Attention-Triplet’ provides sufficient details that how the attention modules work. The manuscript highlights their role in enhancing feature extraction and refining focus on relevant regions, providing a clear and systematic progression.

2. Ensemble Learning Strategy: Sections 3.6 and 3.7 are dedicated to describing the $Chi^2$ Weighted Ensemble (CWE) method and its Multi-Layer extension in detail. These sections explain how CWE differs from traditional ensemble methods and its contribution to achieving superior model performance. Figures 7 and 8 visually support this explanation, making the approach easy to understand.

3. Clarity and Cohesion: We have already ensured that the methodology section presents a cohesive flow from inception to evaluation. Furthermore, the ablation study in Section 4.5 demonstrates the contributions of each component to the overall results, establishing their combined effectiveness in improving accuracy and interpretability.

We believe that these sections adequately address the concerns regarding clarity, novelty, and methodology.________________________________________

To Reviewer 2

Reviewer#2, Comment # 1: The manuscript titled " Chi2 Weighted Ensemble: A Multi-Layer Ensemble Approach for Skin Lesion Classification using A Novel Framework - Optimized RegNet Synergy with Attention-Triplet” is found to be well-structured. However, the manuscript needs to improve significantly. However, I have suggested revision that I believe will further enhance the clarity and impact of your manuscript.

Author response: We sincerely thank the reviewer for this pleasant observation and providing this insightful comment.

Reviewer#2, Comment # 2: The abstract section needs to be updated. I suggest highlighting specific contributions or insights gained from the research in the abstract section. This will help differentiate your study from existing literature reviews on the topic and emphasize its novelty.

Author response: We appreciate the reviewer for their insightful suggestion.

Author action: We have updated the abstract to highlight the specific contributions and insights gained from our research. This revision emphasizes the novelty of our approach, differentiating it from existing studies and underscoring the unique aspects of our work.

Reviewer#2, Comment # 3: The quality of the figures needs to be improved in the revised manuscript.

Author response: Thank you for your feedback regarding the quality of the figures.

Author action: We have used higher-quality images in the revised manuscript. However, due to the PLOS ONE guidelines, we are required to submit figures in TIFF format, which can sometimes result in a slight reduction in resolution. Despite this, we have ensured that the images are of the best possible quality within the format constraints. We hope this addresses your concern.

Reviewer#2, Comment # 4: Arrange significant figures throughout the manuscript.

Author response: Thank you for your suggestion regarding the arrangement of significant figures throughout the manuscript.

Author action: In response, we have added an additional figure to enhance the clarity of the GradCAM process. Additionally, we have tried to reorganize the figures, if possible, for better coherence and to ensure they are presented in a more logical and convenient order. We believe these changes improve the overall flow and understanding of the manuscript.

Reviewer#2, Comment # 5: I recommend incorporating the recent papers that have been discussed and cited in the manuscript.

Author response: Thanks for the insightful suggestions to refine the literature review more clearly.

Author action: We have incorporated six additional recent studies into the manuscript, including one from 2024 and five from 2023, to ensure that the most current research is represented and strengthens the foundation of our work.

Reviewer#2, Comment # 5: Manuscript briefly elaborated on specific research gaps or areas that require further investigation. This could help readers understand the potential directions for future studies.

Author response: Thanks to the reviewer for such insightful observation.

Author action: In response, we have added a dedicated ‘Section 6 Future Work and Research Directions’ in page 41’ outlining specific research gaps and areas that require further investigation. This section highlights the potential directions for future studies, providing readers with a clearer understanding of the next steps and opportunities for advancing this research.

Reviewer#2, Comment # 6: Please add a future direction section or include it with the conclusion.

Author response: Thanks to the reviewer for such insightful advice.

Author action: We have included the future directions in the ‘Section 6 Future Work and Research Directions’ in page 41, as suggested, to provide a clear outline of potential areas for further research and development.

Reviewer#2, Comment # 7: Please make sure to define each acronym at its first use. Check through the entire manuscript to make sure it is defined at the first use.

Author response: Thank you for highlighting the importance of defining acronyms at their first use.

Author action: We have thoroughly reviewed the entire manuscript to ensure that all acronyms are clearly defined upon their initial mention. We are confident that this has been addressed comprehensively throughout the text.

Reviewer#2, Comment # 8: The English of the manuscript needs to be improved.

Author response: Thank you for your comment regarding the language quality of the manuscript.

Author action: We have carefully rechecked the entire manuscript for grammatical errors and typos, and have made revisions to improve the overall language structure. We have also refined the clarity and flow of the text to ensure better readability and precision. We hope these improvements enhance the quality of the manuscript.

Reviewer#2, Comment # 9: The conclusion section should be revised and concise.

Author response: Thanks to the reviewer for such insightful observation.

Author action: We have revised the conclusion section to make it more concise while retaining the key points of our study.

To Reviewer 3

Reviewer#3, Comment # 1: Dr. Anwar Hossain Efat's manuscript describes the Chi2 Weighted Ensemble: A multi-layer ensemble approach for skin Lesion classification using A novel framework- optimized regnet synergy with attention-triplet. This approach has been reported in the literature several times, and regular studies have been conducted on Trans

---

## [Decision Letter · Decision Letter 1]

12 Mar 2025

Chi2 Weighted Ensemble: A Multi-Layer Ensemble Approach for Skin Lesion Classification using A Novel Framework - Optimized RegNet Synergy with Attention-Triplet

PONE-D-24-37946R1

Dear Dr. Efat,

We’re pleased to inform you that your manuscript has been judged scientifically suitable for publication and will be formally accepted for publication once it meets all outstanding technical requirements.

Kind regards,

Fatih Uysal, Ph.D.

Academic Editor

PLOS ONE

Additional Editor Comments (optional):

After considering the referees' comments and evaluating the quality of the paper, it has been decided to accept it due to its potential to contribute to the literature and its final form.

Reviewers' comments:

Reviewer's Responses to Questions

**Comments to the Author**

1. If the authors have adequately addressed your comments raised in a previous round of review and you feel that this manuscript is now acceptable for publication, you may indicate that here to bypass the “Comments to the Author” section, enter your conflict of interest statement in the “Confidential to Editor” section, and submit your "Accept" recommendation.

Reviewer #3: All comments have been addressed

Reviewer #4: (No Response)

2. Is the manuscript technically sound, and do the data support the conclusions?

Reviewer #3: Yes

Reviewer #4: Yes

3. Has the statistical analysis been performed appropriately and rigorously? 

Reviewer #3: Yes

Reviewer #4: I Don't Know

4. Have the authors made all data underlying the findings in their manuscript fully available?

Reviewer #3: Yes

Reviewer #4: (No Response)

5. Is the manuscript presented in an intelligible fashion and written in standard English?

Reviewer #3: Yes

Reviewer #4: Yes

6. Review Comments to the Author

Reviewer #3: The authors acknowledge all the comments raised during the revision, and therefore the manuscript can be accepted in its present form.

Reviewer #4: I read the manuscript and the answers of the authors to the other reviewers. The manuscript is suitable for the publication.

7. PLOS authors have the option to publish the peer review history of their article (what does this mean?). If published, this will include your full peer review and any attached files.

Reviewer #3: **Yes: **THUMPATI PRASANTH

Reviewer #4: No

---

## [Editor Report · Acceptance letter]

PONE-D-24-37946R1

PLOS ONE

Dear Dr. Efat,

I'm pleased to inform you that your manuscript has been deemed suitable for publication in PLOS ONE. Congratulations! Your manuscript is now being handed over to our production team.

Kind regards,

on behalf of

Dr. PLOS Manuscript Reassignment

Staff Editor

PLOS ONE